# Haploinsufficiency of autism spectrum disorder candidate gene NUAK1 impairs cortical development and behavior in mice

Virginie Courchet[1], Amanda J. Roberts[2], Géraldine Meyer-Dilhet[1], Peggy Del Carmine[1], Tommy L. Lewis Jr [3], Franck Polleux [3] & Julien Courchet [1]

Recently, numerous rare de novo mutations have been identified in patients diagnosed with autism spectrum disorders (ASD). However, despite the predicted loss-of-function nature of some of these de novo mutations, the affected individuals are heterozygous carriers, which would suggest that most of these candidate genes are haploinsufficient and/or lead to expression of dominant-negative forms of the protein. Here, we tested this hypothesis with the candidate ASD gene *Nuak1* that we previously identified for its role in the development of cortical connectivity. We report that *Nuak1* is haploinsufficient in mice with regard to its function in cortical development. Furthermore *Nuak1*$^{+/-}$ mice show a combination of abnormal behavioral traits ranging from defective spatial memory consolidation, defects in social novelty (but not social preference) and abnormal sensorimotor gating. Overall, our results demonstrate that *Nuak1* haploinsufficiency leads to defects in the development of cortical connectivity and a complex array of behavorial deficits.

[1] Univ Lyon, Université Claude Bernard Lyon 1, CNRS UMR-5310, INSERM U-1217, Institut NeuroMyoGène, F-69008 Lyon, France. [2] Department of Neurosciences, The Scripps Research Institute, La Jolla, CA 92037, USA. [3] Department of Neuroscience, Zuckerman Mind Brain Behavior Institute and Kavli Institute for Brain Science, Columbia University, New York, NY 10032, USA. Correspondence and requests for materials should be addressed to F.P. (email: fp2304@cumc.columbia.edu) or to J.C. (email: julien.courchet@univ-lyon1.fr)

Neurodevelopmental disorders form a group of disabilities resulting from impairment of brain development and include syndromes such as Autism Spectrum Disorders (ASD), intellectual disability, Attention Deficit/Hyperactivity Disorder (AD/HD), and schizophrenia. The etiology of neurodevelopmental disorders is complex as it involves a marked genetic contribution but probably also some environmental factors and/or a complex interaction between genes and environment[1–4]. Despite strong heritability, less than 10% of syndromes such as ASD can be attributed to dominant mutations in a single gene[1]. Recent studies revealed a large array of rare, de novo mutations in ASD patients[5,6]. Yet their biological significance remains elusive largely because, despite the predicted loss-of-function nature of some of these de novo mutations, the affected individuals are heterozygous carriers, suggesting that most of these candidate genes are either haploinsufficient and/or that these mutations lead to expression of dominant-interfering forms.

The cellular and molecular mechanisms underlying the establishment of neuronal connectivity have been extensively studied. Extracellular factors such as axon guidance cues, neurotrophins, and other growth factors convey information provided by the local environment that is relayed to intracellular signaling molecules to coordinate axon morphogenesis through the triggering of local cellular processes[7]. One such signaling relay, the serine/threonine kinase LKB1, is involved in several aspects of axonal development in the mouse cortex. Lkb1 is a tumor suppressor gene encoding a "master kinase" that activates a group of 14 downstream effectors forming a specific branch of the kinome and related to the metabolic regulator AMPK[8,9]. LKB1 exerts sequential functions in neuronal development through the activation of at least two distinct sets of AMPK-related kinases (AMPK-RK): neuronal polarization and axon specification through the kinases BRSK1/2 (SAD-A/B)[10–12], and axon elongation and branching through NUAK1 (ARK5/OMPHK1)[13]. Furthermore the BRSK ortholog SAD-1 is involved in synapse formation in C. elegans[14,15], whereas in the mouse cortex LKB1 has been shown to regulate synaptic transmission and neurotransmitter release through its ability to control presynaptic mitochondrial $Ca^{2+}$ uptake[16].

Several genes belonging to the AMPK-RK family came out of recent unbiased genetic screens for rare de novo mutations associated with neurodevelopmental disorders (Supplementary Table 1)[17–19]. Yet the functional consequences of these mutations and to what extent they contribute to the physiopathology of the disease remain unknown. To address this question, we focused on Nuak1, one of the AMPK-RK genes strongly expressed in the developing brain[13,20]. Mutations in the Nuak1 gene have been linked to ASD[17,21], cognitive impairment[19], or AD/HD[22]. Interestingly the patients carrying these rare de novo mutations are heterozygous carriers. Therefore, the actual contribution of these mutations to the disease mechanisms suggests that Nuak1 must be either haploinsufficient and/or the mutated allele must be exerting a dominant-negative influence over the wild-type allele.

In this study, we present evidence demonstrating that the inactivation of one allele of the Nuak1 gene results in a complex phenotype including growth retardation, defective axonal mitochondria trafficking, and decreased terminal axon branching of cortico-cortical projections. We reveal that Nuak1 heterozygosity leads to disruption of several cognitive processes, alters the preference for social novelty without affecting conspecific social preference, and decreases the prepulse inhibition (PPI) of the acoustic startle reflex. Finally, using a gene-replacement strategy we characterized the functional consequence of Nuak1 rare de novo mutation identified in ASD patients.

Overall, our results indicate that Nuak1 is haploinsufficient regarding its role in the development of cortical connectivity and

support the hypothesis that loss-of-function mutations in Nuak1 participates in the etiology of neurodevelopmental disorders.

## Results

**NUAK1 haploinsufficiency impairs mouse cortical development.** Using in situ hybridization, we found that Nuak1 mRNA is highly expressed in the developing and adult mouse brain, with specific expression pattern in the developing and adult neocortex, pyriform cortex, hippocampus, and cerebellum. In contrast, Nuak1 expression was low or absent in the dorsal thalamus and striatum (Supplementary Fig. 1A-I). Western-blot analyses from mouse brain extracts confirmed that NUAK1 protein is expressed at high level in the developing and adult cortex. In the hippocampus, NUAK1 expression was maintained throughout all developmental stages, although it decreased with age (Supplementary Fig. 1J).

In mice, constitutive knockout (KO) of Nuak1 is lethal in the late embryonic/perinatal period due to an abdominal midline closure defect called omphalocoele[20]. Despite strong expression in the embryonic brain, the initial characterization of Nuak1 KO embryos did not reveal any obvious morphological brain defect[20]. Accordingly, we did not observe any difference between wild type (WT), heterozygous (HET), and KO embryos in axonal markers (SMI312) or cortical lamination (TBR1: layer 6, and CTIP2: layer 5), indicating no major abnormalities in neuronal polarization, axon tract formation, neurogenesis, neuronal migration, and lamination (Supplementary Fig. 2A-F). NUAK1 protein level was markedly decreased (~50%) in HET neurons (Fig. 1a), suggesting that the remaining allele does not compensate for the loss of one copy of Nuak1. We observed no change in the expression of several members of the AMPK family including NUAK2, AMPK, and BRSK2 (SAD-A) and a moderate increase in BRSK1 (SAD-B) expression resulting from the decrease of NUAK1 expression (Supplementary Fig. 2M), suggesting limited compensation of Nuak1 loss by other AMPK-like family members, matching our previous observations[13] that NUAK1 exerts a specific function in developing cortical neurons.

Unlike homozygotes constitutive knockout (NUAK1$^{-/-}$), heterozygous mice (NUAK1$^{+/-}$) are viable and fertile and do not display the omphalocoele phenotype. Quantitative analysis revealed a moderate but significant growth retardation in NUAK1$^{+/-}$ compared to WT littermates (Fig. 1b, c), a phenotype more pronounced in males than females. We performed histochemical analyses of NUAK1$^{+/-}$ mouse brains at Postnatal day (P)21 (weaning) (Fig. 1d–e) or P40 (young adult) (Supplementary Fig. 2G-L). The global organization of the brain appeared largely normal, with no significant difference in cortical lamination (Supplementary Fig. 2I-L) between WT and HET animals. At the cellular level, neuronal differentiation also seemed unaffected with normal segregation of somatodendritic and axonal markers (Supplementary Fig. 2G-H). Yet the most striking observation was a marked enlargement of the lateral ventricles in NUAK1$^{+/-}$ mice never observed in WT littermates on this C57Bl/6 J genetic background (Fig. 1d–e and Supplementary Fig. 2g–h). The size of lateral ventricles was nearly doubled in NUAK1$^{+/-}$ mice (Fig. 1f), which was accompanied by a corresponding thinning of the cortex along its radial axis (Fig. 1g), whereas the volume of two other telencephalic brain structures (the striatum and hippocampus) was not affected (Fig. 1h–i). Overall, our data suggest that although NUAK1$^{+/-}$ mice are viable, Nuak1 is haploinsufficient with regards to its function during brain development.

**Dose-dependent role of NUAK1 in cortical axon development.** We previously characterized that NUAK1 is necessary and sufficient for terminal axon branching of layer 2/3 cortical neurons

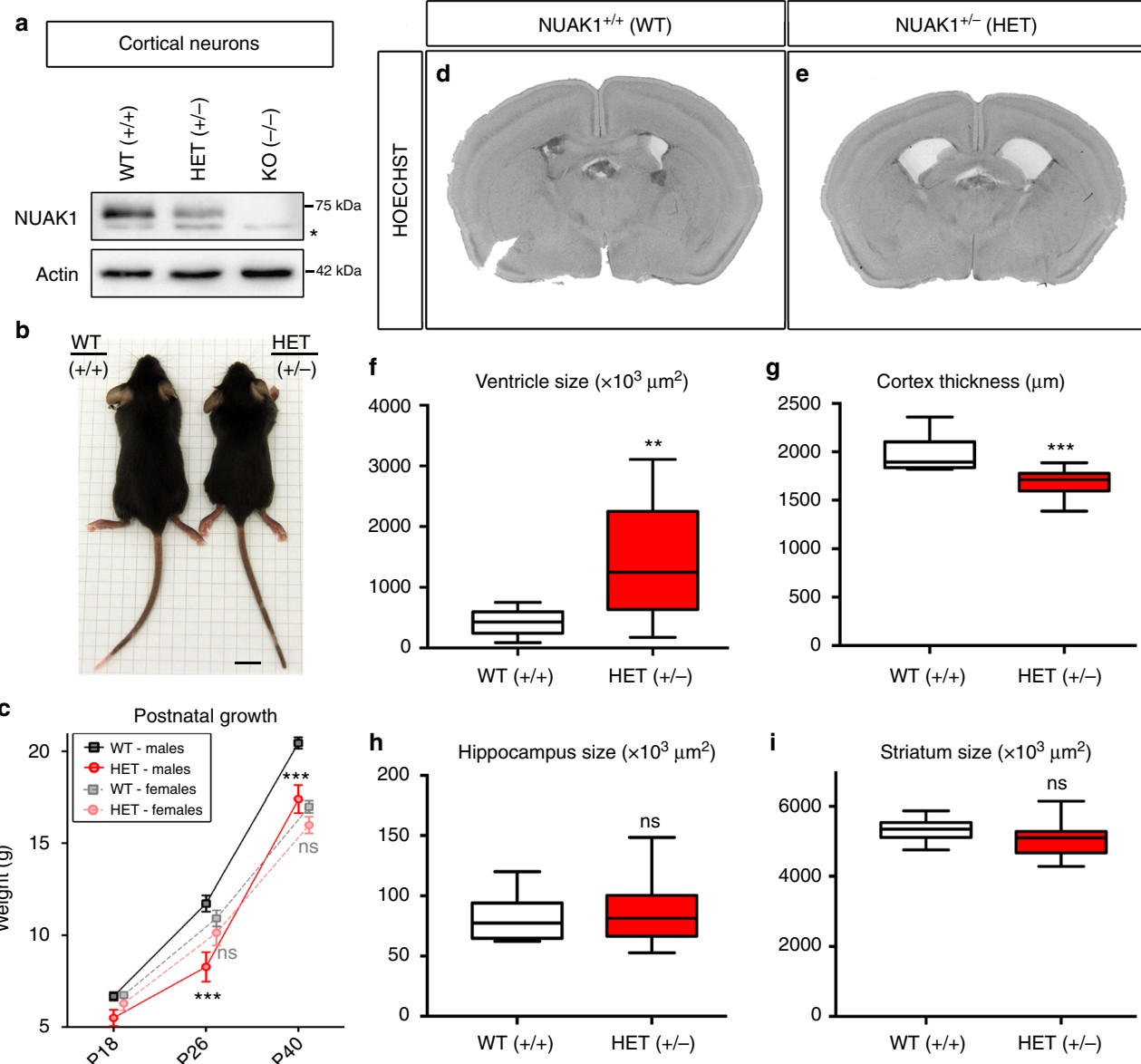

**Fig. 1** Growth retardation and hydrocephaly in NUAK1 mutant mice. **a** Western-blot analysis of NUAK1 expression in 5DIV cortical neurons of WT, HET, and KO animals. The star indicates a non-specific band. **b** Body size difference between males WT and HET NUAK1 mice at age P26. Scale bar 1 cm. **c** Postnatal body weight growth curve of WT and HET NUAK1 mice before (P18), or after (P26-P40) weaning. Average ± SEM. $N_{WT\text{-}females} = 17$, $N_{HET\text{-}females} = 14$, $N_{WT\text{-}males} = 14$, and $N_{HET\text{-}males} = 8$. Analysis: 2-way ANOVA with Bonferroni's multiple comparisons. **d, e** Representative coronal sections of P21 NUAK1 WT (**d**) and HET (**e**) mice stained with Hoechst dye. **f–i** Measurement of ventricle size (**f**), cortex thickness (**g**), hippocampus (**h**), and striatum size (**i**) from P21 coronal sections of NUAK1 WT and HET animals. Data represents min, max, median, 25th, and 75th percentile. Analysis: Two-tailed unpaired T-test. (**f**, **g**, and **i**) $N_{WT} = 10$, $N_{HET} = 15$. **h** $N_{WT} = 7$, $N_{HET} = 13$. ns: $p > 0.05$, **$p < 0.01$, ***$p < 0.001$

through its ability to regulate the presynaptic capture of mitochondria[13]. In order to determine if these phenotypes are also observed upon loss of one copy of *Nuak1*, we first performed neuronal cultures following ex utero cortical electroporation (EUCE) of WT, NUAK1[+/−], and NUAK1[−/−] embryos at E15.5. We observed a significant reduction of axon length and branching in NUAK1[+/−] and NUAK1[−/−] neurons when compared to WT neurons (Fig. 2a–c). The fraction of stationary mitochondria in the axon was markedly decreased for both NUAK1[+/−] and NUAK1[−/−] neurons compared to control WT neurons (Fig. 2d–J). Quantifications confirmed that NUAK1 exerts a dose-dependent effect on axon growth and branching and that HET neurons display a significant reduction in both maximum axon length and the number of collateral branches (Fig. 2g–i).

We subsequently turned to long-term in utero cortical electroporation (IUCE) in order to label the axons of layer 2/3 Pyramidal neurons (PNs) in the primary somatosensory cortex (S1) in WT (NUAK1[+/+]) and HET (NUAK1[+/−]) mice in vivo. Mice were examined at Postnatal day (P)21, when terminal axon branching of these cortico-cortical projecting (callosal) axons is adult-like[23,24]. The loss of one copy of *Nuak1* had no effect on neurogenesis, radial migration, and axon formation (Fig. 2k–l). Furthermore, callosal axons crossed the midline and reached the contralateral hemisphere. However, we observed a marked decrease in terminal axon branching on the contralateral hemisphere in NUAK1[+/−] mice compared to WT littermates (Fig. 2m, n). Interestingly, axon branching on the ipsilateral hemisphere was unaffected (Fig. 2o, p). A similar phenotype was

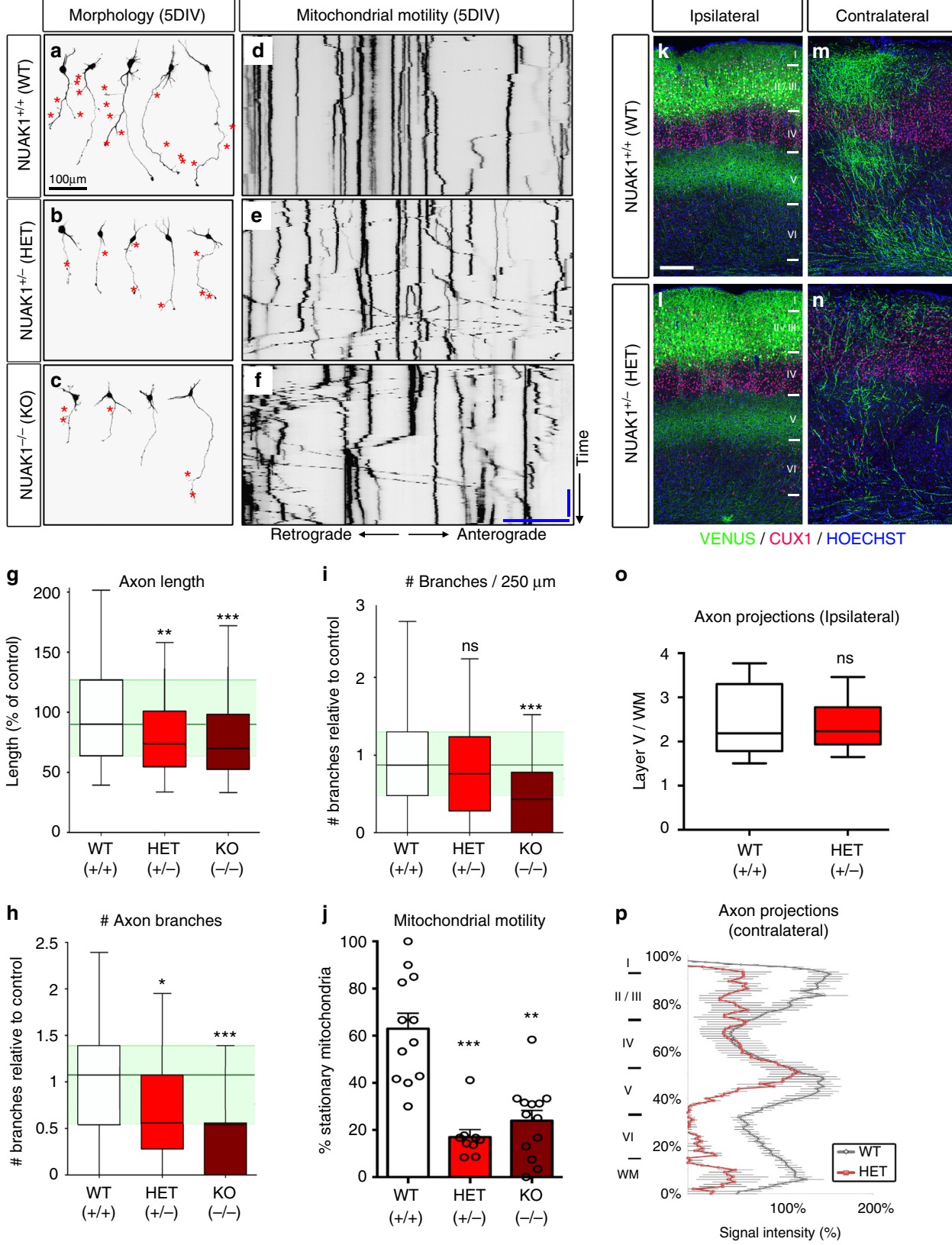

Figure legend elements visible: Morphology (5DIV), Mitochondrial motility (5DIV), Ipsilateral, Contralateral. NUAK1+/+ (WT), NUAK1+/− (HET), NUAK1−/− (KO). 100µm. Time. Retrograde ← → Anterograde. VENUS / CUX1 / HOECHST.

g Axon length; h # Axon branches; i # Branches / 250 µm; j Mitochondrial motility; o Axon projections (Ipsilateral); p Axon projections (contralateral).

observed in older mice (P90) mice (Supplementary Fig. 3), indicating that this deficit in terminal axon branching does not result from a simple delay in axon development and persists through adulthood.

**Alteration of the cognitive functions upon NUAK1 hetero-zygosity.** Despite being involved in cortical axon morphogenesis (present study and ref.[13]), the functional consequences of *Nuak1* heterozygosity have not been explored. In order to investigate the

**Fig. 2** NUAK1 is haploinsufficient for axon development. **a–c** Representative neurons imaged after 5DIV in WT, HET, and KO animals. Neuron morphology was visualized through mVenus expression. Red star (*) points to branch position. **d**, **f** Representative kymographs of axons in WT, HET, and KO neurons at 5DIV. Mitochondria were visualized through expression of mito-DsRed. Blue bar (vertical): 5 min. Blue bar (horizontal): 25 μm. **g**, **i** Quantifications showed a gradual reduction in axon length (**g**) and collateral branches (**h**, **i**) in HET and KO animals. Data represents min, max, median, 25th, and 75th percentile (**g–i**) or average ± SEM (**j**). $N_{WT} = 127$, $N_{HET} = 232$, $N_{KO} = 203$. Analysis: one way ANOVA with Bonferroni's post test. ns: $p > 0.05$, *$p < 0.05$, **$p < 0.01$, ***$p < 0.001$. **j** Quantification of mitochondrial motility in the axon. Average ± SEM. $N_{WT} = 12$, $N_{HET} = 9$, $N_{KO} = 13$. Analysis: Kruskal–Wallis test with Dunn's multiple comparison. **k–n** Higher magnification of the ipsilateral (**k–l**) or contralateral side (**m–n**) of P21 WT or HET mice following in utero cortical electroporation of mVenus. Blue: Hoescht, Red: CUX1 immunostaining. Scale bar: 250 μm. **o–p** Quantification of normalized mVenus fluorescence in Layer 5 of the ipsilateral cortex (min, max, median, 25th, and 75th percentile) (**o**) and along a radial axis in the contralateral cortex (Average ± SEM) (**p**). $N_{WT} = 6$, $N_{HET} = 5$. Analysis: two-tailed unpaired $T$-test (**o**). ns: $p > 0.05$

effect of *Nuak1* haploinsufficiency on mouse behavior, we generated age-matched cohorts of WT and NUAK1[+/−] mice and performed a battery of behavioral assays relevant to neurodevelopmental disorders. Vision and olfaction were not impaired in NUAK1[+/−] mice, as demonstrated by tracking of head movement using an optomotor test (Supplementary Fig. 4A) and by the habituation/dishabituation to novel odor test (Supplementary Fig. 4B). Furthermore NUAK1[+/−] mice exhibited no difference in spontaneous locomotor activity and exploratory behavior in the Open Field (Supplementary Fig. 5A-C).

Next, we tested how NUAK1 heterozygosity affects memory formation and consolidation through three distinct cognitive paradigms: first, we investigated non-spatial memory using the Novel Object Recognition (NOR) task[25] (Fig. 3a). There were no significant sex or genotype effects in the subsequent test trial; however, both WT ($p < 0.005$) and NUAK1[+/−] ($p < 0.01$) mice showed significant increase in number of object contacts (F $(1,35) = 14.21$, $p < 0.001$) and contact times (F$(1,35) = 15.63$, $p < 0.001$), suggesting normal levels of interest for the novel objects in both genotypes (Fig. 3b, c).

Second, we tested spatial memory formation and maintenance using the Barnes Maze assay[26] (Fig. 3c). While this test is similar to the Morris Water Maze in probing spatial memory, it is less physically taxing and does not induce stress/anxiety response linked to swimming in water[27]. There were no significant differences involving sex or genotype in the acquisition of escape behavior. In the probe trial both WT and NUAK1[+/−] mice showed a clear preference for the target quadrant relative to other quadrants (Fig. 3d, probe test #1 [F$(1,44) = 11.1$, $p < 0.005$]). However, when retested in the same conditions after one month with the same platform location, while there was a significant effect of quadrant exploration time (F$(1,44) = 6.9$, $p < 0.01$), only WT mice showed a significant preference for the target quadrant ($p = 0.005$), while NUAK1[+/−] mice did not ($p = 0.45$; Fig. 3e, probe test #2).

Finally, we assessed associative learning and short-term memory formation using the Fear Conditioning (FC) assay. On day 1 of the test, adult NUAK1[+/−] and WT littermates were placed in the experimental apparatus and their spontaneous activity was recorded and automatically measured (Fig. 3g, day 1). After a 2 min period, a neutral conditioned stimulus (30 s auditory tone) was paired with a single aversive unconditioned stimulus (mild electric foot shock). There was no difference in the spontaneous activity of WT and NUAK1[+/−] mice during this phase (Fig. 3g, day 1). On the second day, contextual and cued fear conditioning was assessed over 3 minutes trials. We observed a trend, albeit not significant, toward a reduction of the startle response during the contextual phase of the assay both for the total duration ($p = 0.23$; Fig. 3h, left) and number of startle events ($p = 0.67$; Fig. 3h, right). On the contrary, NUAK1[+/−] mice had a marked reduction of the cue-induced fear response both for total duration ($p = 0.012$; Fig. 3h, left) and number of events recorded ($p = 0.039$; Fig. 3i, right). Altogether, these results indicate that

NUAK1 partial loss-of-function disrupts short and long-term memory processes in the mouse.

**NUAK1 heterozygosity impairs novelty preference and sensory gating**. We next performed a series of behavioral assays relevant to autistic-like traits. We observed spontaneous activity through Open Field sessions (Supplementary Fig. 5A-F) but did not detect any differences between WT and NUAK1[+/−] mice in behaviors classically associated with anxiety or repetitive behavior. Since these behaviors might be altered by the stress induced by the open field environment, we repeated these observations in a home-cage environment. Similarly, there was no genotype-dependent difference in spontaneous behaviors (grooming, scratching, or rearing), although we observed a mild increase in spontaneous activity and digging (Supplementary Fig. 5G-J).

Social interactions were assessed using the 3-chamber sociability assay (Fig. 4a)[28]. There was no overall genotype difference, but a significant overall effect of time in chamber (F$(2,132) = 227.6$, $p < 0.0001$) and direct contact times (F$(1,66) = 51.3$, $p < 0.0001$). NUAK1[+/−] mice displayed the same preference, measured by the time spent in each chamber (Fig. 4b) and direct object/mouse contact time (Fig. 4c) as their WT littermates when offered a choice between another mouse and an inanimate object, indicating that conspecific sociability is not affected by NUAK1 heterozygosity. In a second test, we assessed the preference of NUAK1[+/−] mice for social novelty (Fig. 4d). In this case, there were overall effects of chamber and contact times, but also significant interactions between these factors and genotype (F's – 3.7–5.1, $p < 0.05$). As expected, WT mice spent more time in the chamber with the novel mouse and interacted more with the novel individual. However, NUAK1[+/−] mice displayed no preference between the familiar and the novel mouse (Fig. 4e, f). Importantly, this effect was not due to a lack of interest for novelty since NUAK1[+/−] mice showed a preference for a novel object similar to their WT littermates in the NOR task (Fig. 3a–c). Altogether, our results point to a deficit in the preference for social novelty upon loss of one copy of *Nuak1*.

We subsequently investigated sensorimotor gating through prepulse inhibition (PPI) of the startle response induced by weaker prestimuli[29,30] (Fig. 4g). We observed a marked enhancement of the startle response in NUAK1[+/−] mice ([F $(1,44) = 24.9$, $p < 0.0001$], Fig. 4h). Interestingly, there was a significant interaction between startle sound level x genotype x sex (F$(7,308) = 2.09$, $p < 0.05$), revealing a more pronounced increase in NUAK1[+/−] males compared to NUAK1[+/−] females (Supplementary Fig. 6B-C). This increased startle response was accompanied by a significant reduction of PPI ([F$(1,44) = 11.5$, $p = 0.002$], Fig. 4i), with stronger effect in males than in females (Supplementary Fig. 6B-C). We retested the mice by submitting NUAK1[+/−] to a milder tone in order to normalize the intensity of the startle response between WT and NUAK1[+/−] animals. Under these conditions, we observed a milder, and non-significant PPI reduction in NUAK1[+/−] males and no difference

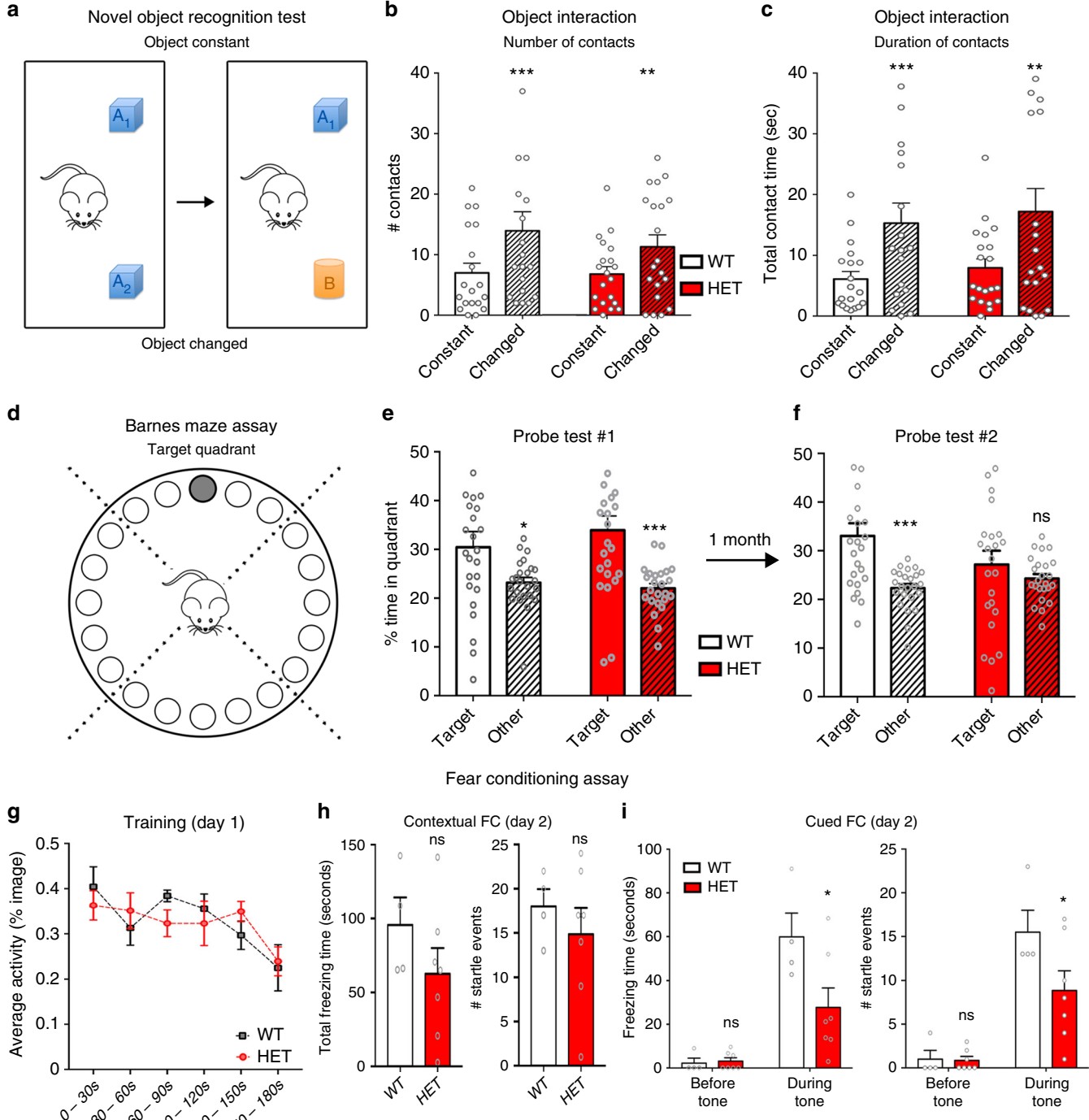

**Fig. 3** Long-term memory consolidation phenotype of NUAK1 HET mice. **a** Novel object assay design. Mice were exposed to two similar objects, followed by a change in one of the two objects. **b**, **c** Quantification of the number (**b**) and duration (**c**) of contacts during the novel object recognition assay. Average ± SEM. $N_{WT} = 19$, $N_{HET} = 20$. Analysis: 2-way ANOVA with Bonferroni's multiple comparisons. **p < 0.01, ***p < 0.001. **d** Barnes Maze spatial memory assay. **e** Quantification of time spent in target quadrant versus in other quadrants. **f** Mice were tested after 1 month to assess memory consolidation. Average ± SEM. Analysis: 2-way ANOVA with Bonferroni's multiple comparisons. ns: p > 0.05, *p < 0.05, ***p < 0.001. **g**, **i** Fear conditioning assay. **g** On day 1, mice were placed in the apparatus for 3 min. After a 2 min period, a 30 s tone was produced, and paired with a single electric shock. Spontaneous activity was recorded over 30 s bins. **h**–**i** on day 2, mice were tested for startle response (total duration and number of events) upon exposition to the same environment (contextual memory) and upon exposure to the tone in a modified environment (cued memory). Average ± SEM. $N_{WT} = 4$, $N_{HET} = 7$. Analysis: Mann–Whitney (**h**) or 2-way ANOVA with Bonferroni's multiple comparisons (**i**). ns: p > 0.05, *p < 0.05

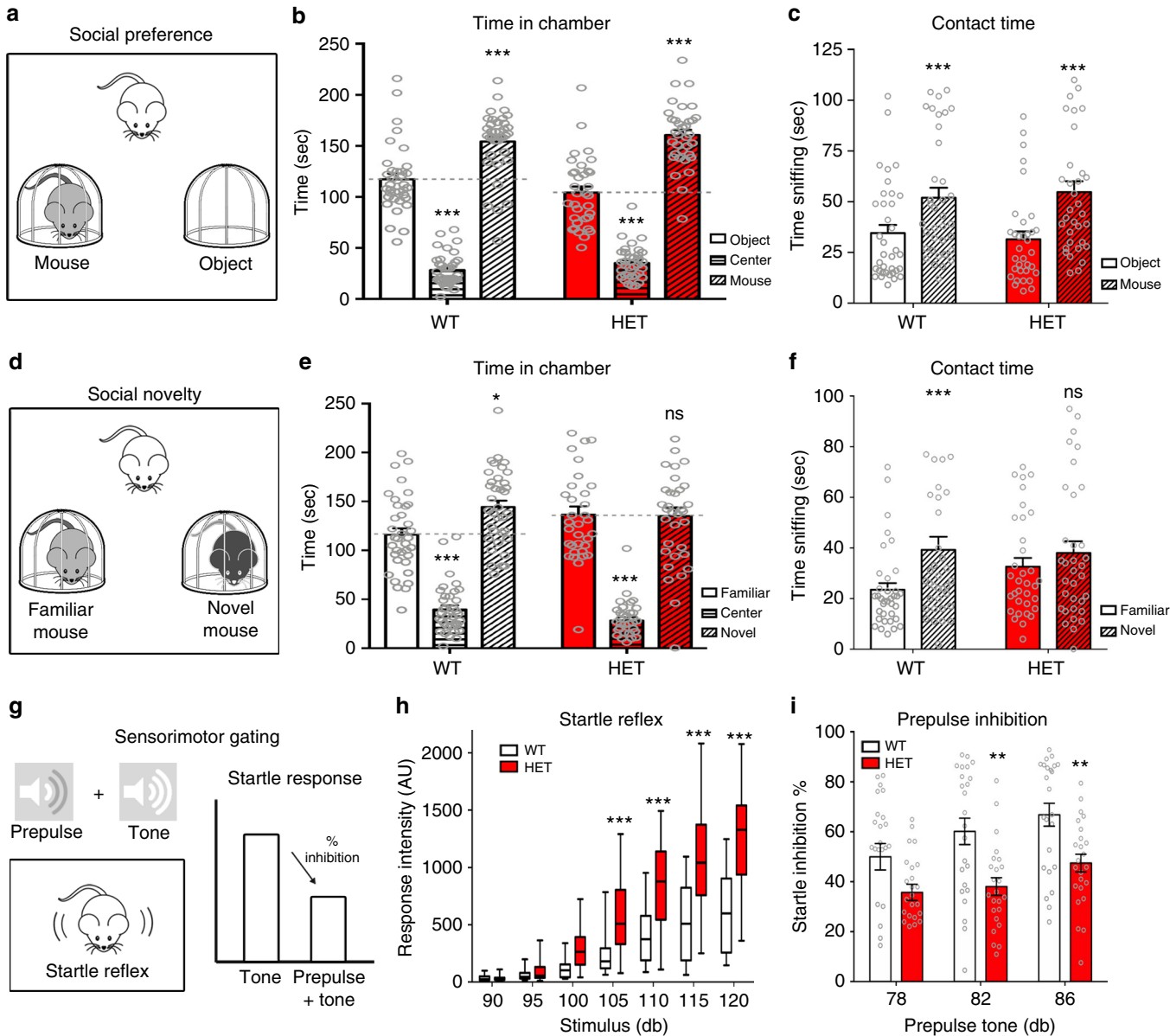

**Fig. 4** Sociability and sensorimotor phenotype of NUAK1 HET mice. **a** 3-chamber sociability assay design. **b, c** Quantification of time spent in each chamber (**b**) and contact time with the mouse or the empty cage (**c**) did not reveal any social deficit in NUAK1 HET mice. Data: Average ± SEM. $N_{WT} = 36$, $N_{HET} = 34$. Analysis: 2-way ANOVA with Bonferroni's multiple comparisons. ***$p < 0.001$. **d** Social novelty assay design. **e, f** Quantification of time spent in chambers (**e**) and contact time with a familiar and a novel mouse (**f**). Data: Average ± SEM. $N_{WT} = 36$, $N_{HET} = 34$. Analysis: 2-way ANOVA with Bonferroni's multiple comparisons. ns: $p > 0.05$, *$p < 0.05$, ***$p < 0.001$. **g** Sensorimotor gating assay and measurement of the startle response. The startle reflex intensity (**h**) and % inhibition of the startle response upon pre-stimulation with a milder tone (**i**) showed altered response in NUAK1 HET. Data represents min, max, median, 25th, and 75th percentile (**h**) or average ± SEM (**i**). $N_{WT} = 24$, $N_{HET} = 24$. Analysis: 2-way ANOVA with Bonferroni's multiple comparisons. **$p < 0.01$, ***$p < 0.001$

in NUAK1$^{+/-}$ females compared to control WT littermates (Supplementary Fig. 6D-E, see figure legend for statistical results). Finally, the optimal InterStimulus Interval (ISI) was similar between WT and NUAK1$^{+/-}$ mice, and the PPI was impaired at all ISI examined, indicating that the deficit in PPI does not result from an abnormality in the temporal function of the startle reflex (Supplementary Fig. 6A). Taken together, our results indicate that NUAK1 heterozygosity induces a deficit in the PPI pointing to a sensory gating defect most likely due to a strong increase in the initial startle response.

**NUAK1 cell-autonomously controls axon branching**. Given the embryonic lethality characterizing the NUAK1$^{-/-}$ constitutive

knockout, we generated a conditional floxed *Nuak1* mouse model (NUAK1$^F$ allele) (Supplementary Fig. 7A-B) in order to compare partial and complete genetic loss of *Nuak1* function during development of cortical connectivity at postnatal stages. We first performed primary cortical neuron cultures coupled with EUCE of plasmids encoding Cre recombinase. The loss of one (NUAK1$^{F/+}$) or two (NUAK1$^{F/F}$) allele of *Nuak1* showed again a dose-dependent effect on axon growth and branching (Supplementary Fig. 7C-H), thus validating this new conditional knockout as a potent loss-of-function allele phenocopying the constitutive knockout.

Next, we performed long-term IUCE of plasmids encoding Cre recombinase in floxed NUAK1 embryos at E15.5 to quantify axon

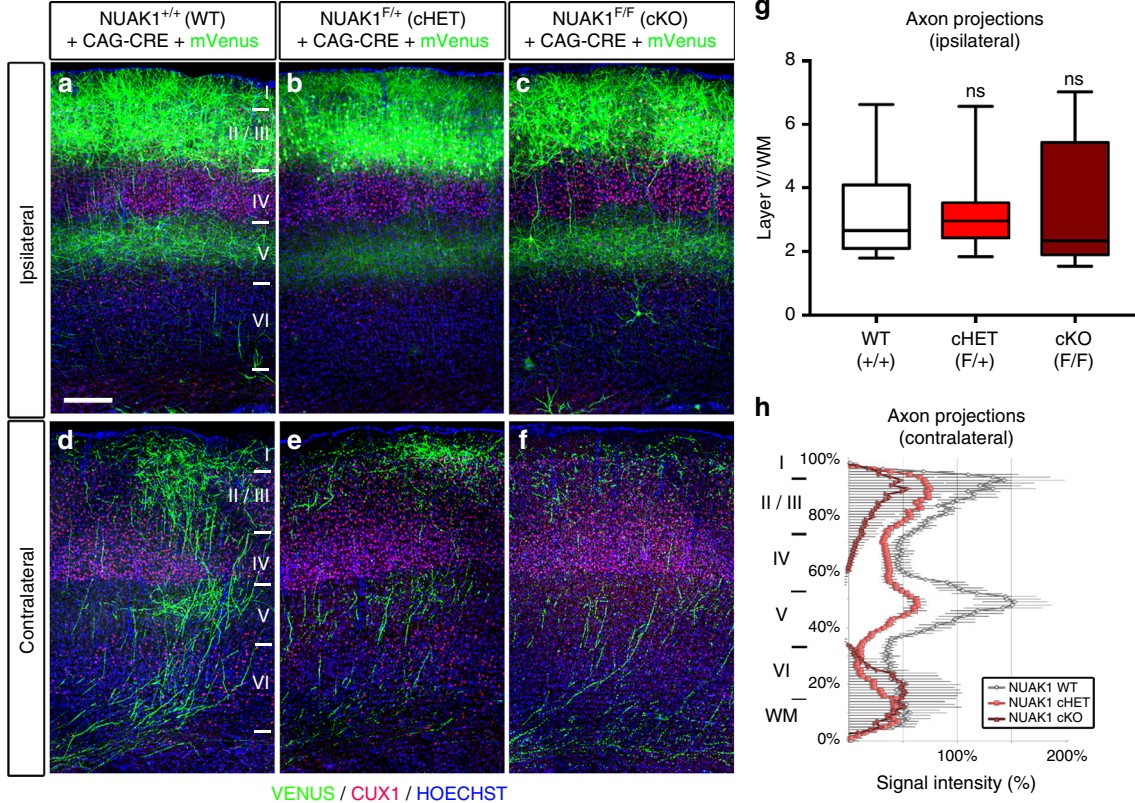

**Fig. 5** Cell autonomous reduction in axon branching of NUAK1-deficient neurons. **a–f** Histochemistry of the ipsilateral or contralateral side of NUAK1$^{+/+}$, NUAK1$^{F/+}$, and NUAK1$^{F/F}$ mice at P21 following in utero electroporation with CRE and the fluorescent protein mVenus. Blue: Hoescht, Red: CUX1 immunostaining. Scale bar: 250 μm. **g–h** Quantification of normalized mVenus fluorescence in Layer 5 of the ipsilateral cortex (**g**) and along a radial axis in the contralateral cortex (**h**). Data: Average ± SEM. $N_{WT} = 6$, $N_{HET} = 13$, $N_{KO} = 6$. Analysis: Mann–Whitney test. ns: $p > 0.05$

branching of layer 2/3 PNs in vivo at P21. Importantly, this approach is also testing the cell-autonomy of NUAK1 deletion since Cre-expressing neurons are relatively sparse (~10% of all layer 2/3 neurons) and their axons grow in a largely wild-type environment. Similarly to our observations in NUAK1$^{+/−}$ mice, Nuak1 conditional deletion had no effect on neurogenesis, radial migration, axon formation and ipsilateral branching (Figs. 5a–c and 5G). However, we observed a marked decrease of contralateral axon branching of layer 2/3 neurons upon inactivation of one or both alleles (Figs. 5d–h), strongly arguing that Nuak1 is haploinsufficient with regard to the establishment of cortico-cortical connectivity during postnatal development.

In order to determine the consequence of a complete deletion of NUAK1 on mouse behavior, we crossed NUAK1$^{F/F}$ mice with NEX$^{CRE}$ mice[31]. The NEX promoter drives CRE expression specifically in intermediate progenitors and in PNs of the dorsal telencephalon, excluding GABAergic interneurons and astrocytes or other non-neuronal cells. Conditional inactivation of one (cHET) or both (cKO) copies of Nuak1 using NEX$^{CRE}$ had no effect on mouse viability and postnatal growth (Fig. 6a and Supplementary Fig. 8A). We then performed the NOR task and the three-chambered social preference and social novelty assay on age-matched cohorts of WT, cHET, and cKO animals. As was the case upon constitutive deletion of NUAK1, there was no overall effect of genotype on novel object recognition (Supplementary Fig. 8B) and conspecific sociability as measured by time in chamber (Fig. 6b) and direct contact times (Supplementary Fig. 8C). However during the social novelty phase of the assay, NUAK1 cHET and NUAK1 cKO mice showed a lack of preference for the novel mouse compared to the familiar mouse

as measured by time in chamber (Fig. 6c) and direct contact times (Supplementary Fig. 8D). Finally, we performed the FC assay on NUAK1 cKO and WT littermate animals. As observed in NUAK1$^{+/−}$ animals, there was no genotype-dependent difference in the spontaneous activity during the training phase of the assay (Fig. 6d) and no statistically different startle response in the contextual FC (Fig. 6e, duration: $p = 0.28$, # startle: $p = 0.39$), however cued-FC was markedly reduced in NUAK1 cKO animals (Fig. 6f, duration: $p = 0.006$, # startle: $p = 0.001$). Taken together our results indicate that the complete inactivation of Nuak1 in PNs of the dorsal telencephalon recapitulates the lack of preference for social novelty and decreased cued fear conditioning observed in NUAK1 HET mice, and thus confirms that NUAK1 haploinsufficiency leads to defects in mouse social behavior.

**Functional analysis of a Nuak1 mutation associated with ASD.** Finally, we tested whether Nuak1 mutations identified in ASD patients affect the function of the protein. The de novo mutation most frequently identified in genetic studies and with the highest confidence score is a nonsense mutation inducing a Premature Termination Codon (PTC) corresponding to Glutamine 433 (mouse 434) of the protein (Fig. 7a and Supplementary Table 1). Notably this PTC mutation occurs in the long, last coding exon of Nuak1 (Supplementary Fig. 7A) and thus is not predicted to be subjected to nonsense-mediated decay[32]. The mutation (hereafter named NUAK1 STOP) leads to the truncation of a large portion of the C-terminal extremity (CTD) of the protein, and removes one activating AKT-phosphorylation site as well as two GILK motifs required for the interaction with the myosin phosphatase complex[33].

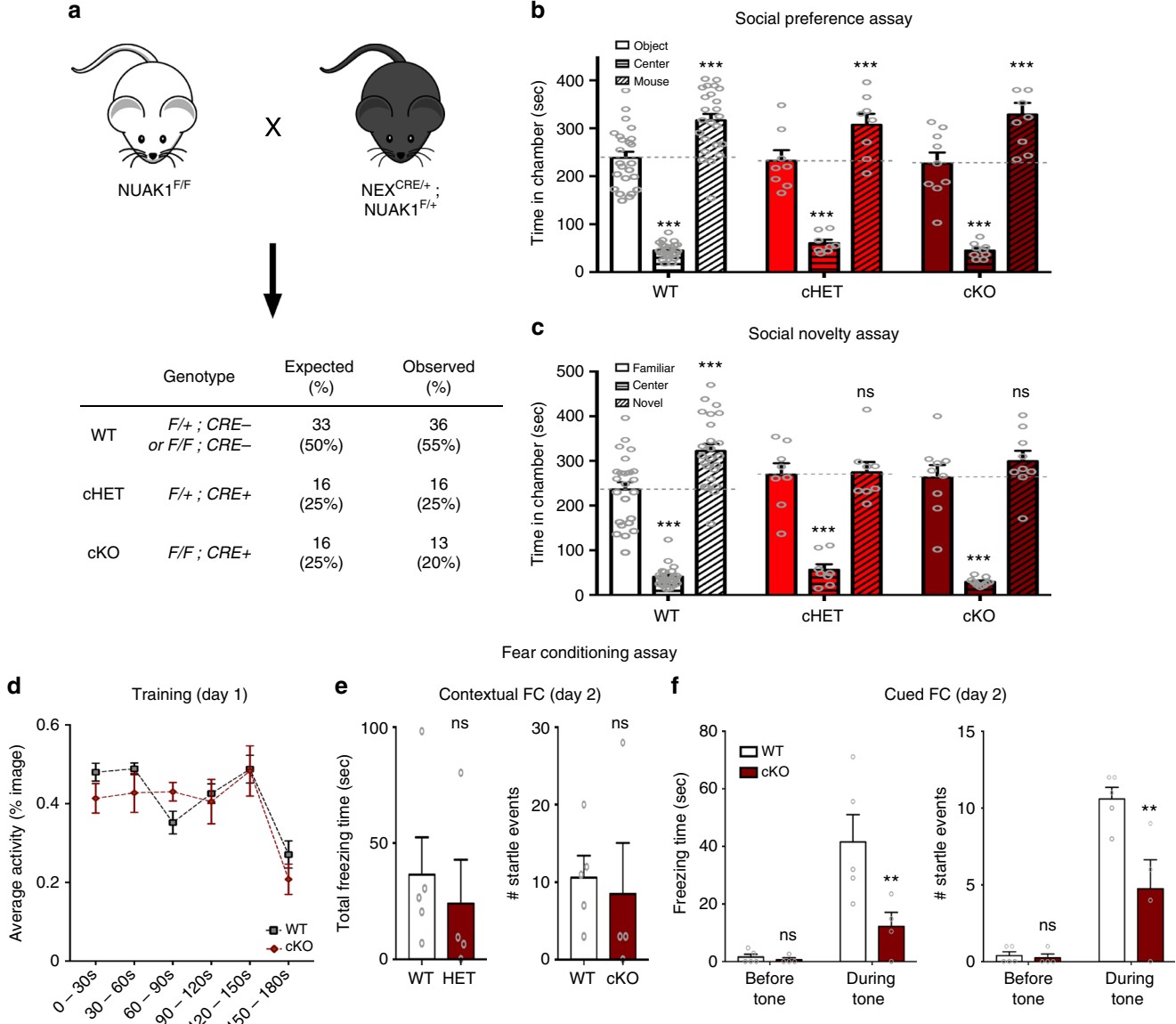

**Fig. 6** Reduced preference for social novelty upon complete loss of NUAK1 in the cortex. **a** Breeding strategy for dorsal-telencephalon specific inactivation of *Nuak1*. Observed proportion of each genotype was matching the expected proportion following mendelian rule. **b**, **c** 3-chamber sociability and social novelty assay in NUAK1 cKO mice. Quantification of time spent in each chamber revealed a decreased preference for social novelty in cHET and cKO mice. Data: Average ± SEM. $N_{WT} = 24$, $N_{cHET} = 8$, $N_{cKO} = 9$. Analysis: 2-way ANOVA with Bonferroni's multiple comparisons. ns: $p > 0.05$, ***$p < 0.001$. **d**–**f** Contextual fear conditioning assay. **d** Automated measurement of spontaneous activity over 30 s bins during the tone-foot shock pairing. **e**, **f** Startle response (total duration and number of events) upon testing of the contextual (**e**) and cued (**f**) fear conditioning. Average ± SEM. $N_{WT} = 5$, $N_{cKO} = 4$. Analysis: Mann–Whitney (**e**) or 2-way ANOVA with Bonferroni's multiple comparisons (**f**). ns: $p > 0.05$, **$p < 0.01$

We used site-directed mutagenesis to introduce this PTC mutation in NUAK1 coding plasmids, creating an expression vector encoding NUAK1 STOP, and used this construct to determine the consequences of NUAK1 protein truncation. We observed by western-blot that the mutation had no impact on the expression of the protein (Fig. 7b), suggesting that the removal of a large portion of NUAK1 CTD is unlikely to reduce protein stability. Importantly, the truncation did not affect the kinase activity of the mutant protein (Fig. 7c).

We then investigated the consequences of expressing this NUAK1 STOP protein on axon growth and branching. To reach that goal, we adopted a gene-replacement strategy to express either WT or mutant NUAK1 STOP protein in NUAK1-null neurons and determined axon morphology at 5DIV. As reported

above upon conditional inactivation of NUAK1 expression through Cre recombinase expression, neurons had shorter axons with reduced branching (Fig. 7d–e). Re-expression of WT NUAK1 protein fully rescued axon length and axon branching (Fig. 7f). On the contrary two mutants, a kinase dead (KD) mutant that totally abolishes NUAK1 catalytic activity, and a mutant of the LKB1-activation site (TA), both failed to rescue the axonal phenotypes (Fig. 7g–h) confirming that NUAK1 function in axon morphogenesis is kinase-dependent. Interestingly, NUAK1 STOP mutant could rescue axon elongation to the same extent as the WT NUAK1, but was not able to rescue collateral branch formation (Fig. 7i–n). Using a similar strategy, we observed that NUAK1 STOP mutant failed to rescue mitochondria immobilization in NUAK1-null neurons (Supplementary

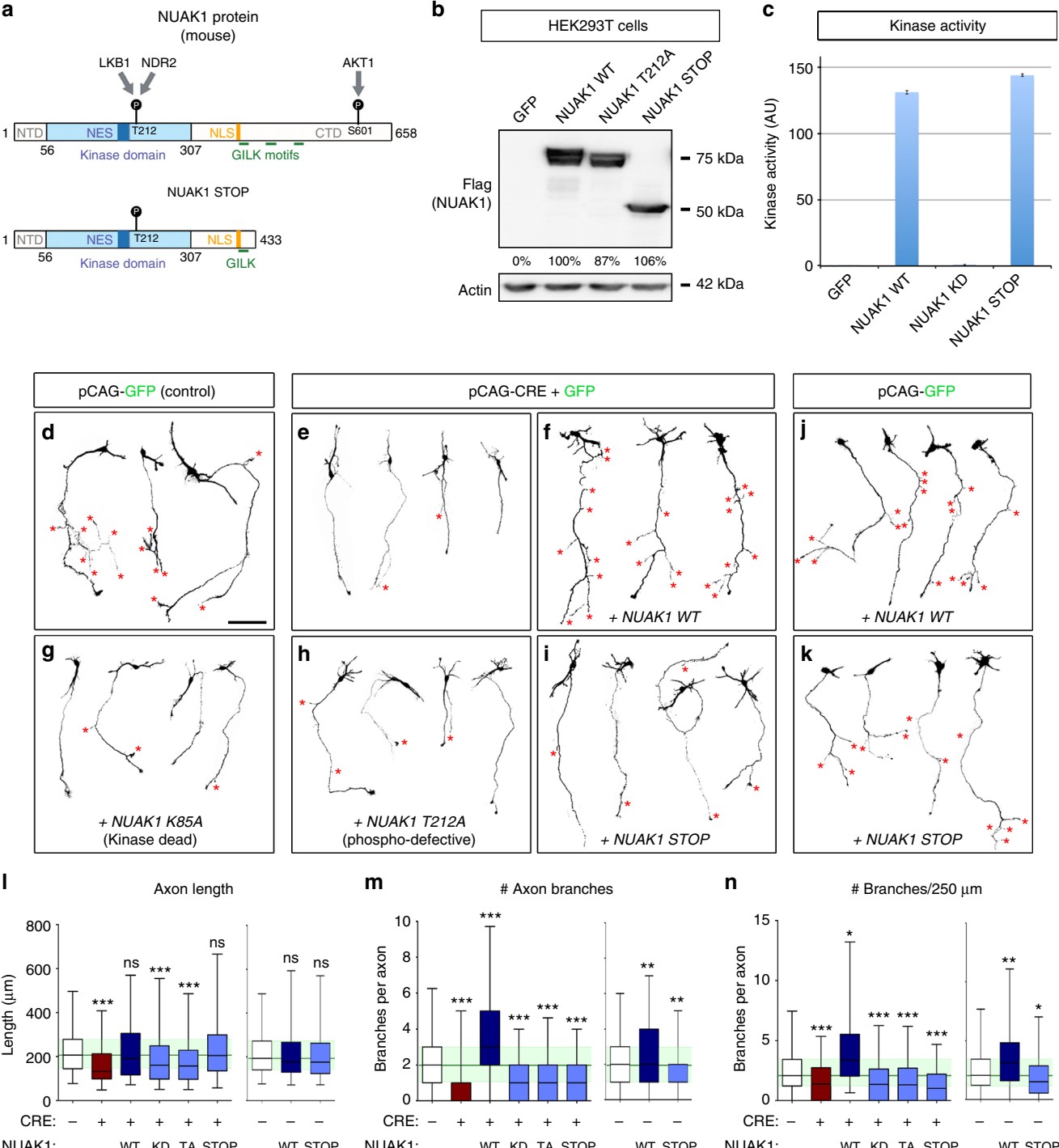

**Fig. 7** Functional analysis of NUAK1 mutations through a protein-replacement strategy. **a** Structure of the NUAK1 protein (mouse) and predicted structure of the Q434STOP mutant. Known or predicted domains are indicated. NTD: N-terminal domain. CTD: C-terminal domain. NES: Nuclear Export Sequence (predicted). NLS: Nuclear Localization Sequence (predicted). **b** Validation of NUAK1 mutant expression in HEK293T cells by western-blot. **c** In vitro measurement of kinase activity of wild type or mutant NUAK1. **d–k** Representative neurons imaged after 5DIV following ex vivo electroporation of the indicated plasmids in NUAK1[F/F] animals. Neuron morphology was visualized through mVenus expression. Red stars (*) points to branch position. Scale bar: 100 μm. **l–n** Quantification of axon length (**l**), number of branches per axon (**m**) and normalized branch number (**n**) in neurons from (**d–k**). Quantifications were normalized to the average value of Control (CTL) condition to allow comparison of distinct, independent experiments. Data represents min, max, median, 25th, and 75th percentile. $N_{CTL} = 189$, $N_{CRE} = 138$, $N_{CRE+WT} = 90$, $N_{CRE+KD} = 150$, $N_{CRE+TA} = 174$, $N_{CRE+STOP} = 125$, $N_{WT} = 134$, $N_{STOP} = 198$. Analysis: Kruskal–Wallis test with Dunn's multiple comparison. ns: $p > 0.05$, *$p < 0.05$, **$p < 0.01$, ***$p < 0.001$

Fig. 9, quantified in Supplementary Fig. 9G). Thus the NUAK1 STOP mutant is a loss-of-function mutation that abolished its function in mitochondria trafficking and axon branching.

We finally investigated whether the NUAK1 STOP mutation acts as a dominant-negative when expressed in neurons expressing endogenous WT NUAK1. We expressed either WT or mutant forms of NUAK1 into WT neurons (NUAK1$^{F/F}$ without Cre) by EUCE and observed axon morphology and mitochondria trafficking at 5DIV. As observed previously[13], overexpression of WT NUAK1 was sufficient to increase axon branching (Fig. 7j) and increase the pool of stationary mitochondria in the axon (Supplementary Fig. 9C). In contrast, neurons expressing the NUAK1 STOP mutant protein had limited to no effect on axon length or branching (Fig. 7k) and mitochondria trafficking (Supplementary Fig. 9E). Quantification of axon morphology confirmed that NUAK1 STOP mutant failed to increase branching (Fig. 7l–n). We only observed a moderate reduction in collateral branch formation upon expression of the NUAK1 STOP mutant. Furthermore, there was no effect of NUAK1 STOP mutant expression on mitochondria trafficking in wild-type neurons (Supplementary Fig. 9G). Altogether, expression of the NUAK1 STOP mutant had only a moderate inhibitory activity over endogenous NUAK1 upon collateral branch formation and no detectable inhibitory effect on axon elongation and mitochondria trafficking, indicating that this mutant behaves mostly as a loss-of-function mutant and phenocopies NUAK1 haploinsufficiency.

## Discussion
In the present study, we characterized for the first time the consequences of *Nuak1* heterozygosity on brain development and cortical circuit formation. Previous observations reported that *Nuak1* is highly enriched in the developing brain[20,34], yet its function in brain development was previously concealed by the embryonic lethality of constitutive *Nuak1* knockout mice. We now report that *Nuak1* expression remains elevated in the mouse brain at various stages of postnatal development. Furthermore our results demonstrate for the first time a dose-dependent effect of NUAK1 on mitochondria trafficking, axon elongation and branching.

The genetic architecture of neurodevelopmental disorders such as ASD is emerging as complex and in the vast majority of cases seems to deviate from mono-allelic, recessive mutations. Recent evidence suggest that more often, rare de novo mutations are found in ASD patients including potential loss-of-function mutations in AMPK-RK such as *Nuak1*[17,19,21], *Sik1*[21], *Brsk1*[18], or *Brsk2*[19]. However, functional assessment of the phenotypic impact of these rare de novo mutations is currently unknown. Here we demonstrate for the first time that *Nuak1* is haploinsufficient with regard to brain development implying most likely that heterozygote loss-of-function mutations in this gene could cause defects in brain connectivity and a range of behavioral defects compatible with ASD, intellectual disability, and schizophrenia.

Many of the AMPK-RKs are enriched in the developing brain and several have been linked to neurodevelopmental disorders. For example, mutations in *Sik1* have been identified in patients suffering from developmental epilepsy[35], and polymorphisms in *Mark1* are associated with an increased risk for ASD[36]. AMPK-RKs might have partially overlapping functions: for example the simultaneous inactivation of the polarity kinases BRSK1 (SAD-B) and BRSK2 (SAD-A) is required to disrupt axon specification in mouse cortical neurons yet *Brsk1;Brsk2* double KO is lethal shortly after birth[12], whereas single gene inactivation is less phenotypic and viable. We previously reported that NUAK1

function in axon branching is not shared by other AMPK-RKs such as AMPK, BRSK1/2, or NUAK2[13]. This lack of redundancy between NUAK1 and other AMPK-RKs in axon branching might explain why even heterozygous mutations in *Nuak1* could have important phenotypic consequences on brain development.

The main phenotype resulting from NUAK1 dosage reduction is a decrease in terminal axon branching of cortico-cortical projections. Unlike *Lkb1* deletion[13], ipsilateral branching of callosal axons was largely unaffected by the loss of *Nuak1*, whereas contralateral branching was reduced upon loss of one (NUAK1$^{+/-}$) or two (NUAK1 cKO) copies of *Nuak1*. Ipsilateral branching can be observed as early as P4 and is generated mostly through interstitial branching[37], contrary to the later terminal contralateral branching. Hence a difference in the timing and/or mechanism of branch formation might explain how ipsilateral and contralateral branching might be differentially affected by *Nuak1* loss. Contralateral axon branching is significantly affected by interfering with neuronal activity[23,38]. For example CUX1 inactivation results in an activity-dependent reduction in contralateral, but not ipsilateral branching of layer 2/3 cortical PNs[39]. Furthermore a disruption of neurotrophin-mediated signaling (BDNF/TrkB) also results in a similar reduction of contralateral axon branching[40] while ipsilateral branching is largely unaffected. Further studies will elucidate how the interplay between neuronal activity and neurotrophin-mediated signaling may differentially and locally regulate axon branching in a NUAK1-dependent way.

In this study, we adopted a gene-replacement strategy to determine the functional relevance of one of the de novo mutations in the *Nuak1* gene potentially to ASD in human[17] leading to the expression of a truncated protein. Importantly, we confirmed in cell lines that the mutant protein is stable, and that the mutation does not abolish its catalytic activity in vitro (Fig. 7), which likely explains how this mutant is able to partially rescue the phenotype upon NUAK1 loss in neurons. Our data suggest that the NUAK1 STOP mutation behaves as a loss-of-function rather than a dominant-negative form of the protein since NUAK1 STOP overexpression in WT cortical neurons did not phenocopy NUAK1 loss-of-function. NUAK1 STOP overexpression in WT cortical PNs had little impact on axon elongation and mitochondria trafficking, and the reduction in axon branching was not as pronounced as the reduction observed upon CRE-mediated NUAK1 inactivation. Interestingly the NUAK1 protein has a long C-terminal tail containing potential regulatory sites including an AKT phosphorylation site[41] and three short GILK motifs promoting NUAK1 interaction with Protein Phosphatase 1β (PP1β)[33]. LKB1 activation is achieved through nuclear export and cytosolic capture within the STRAD/Mo25 complex[42,43]. Similarly it is plausible that NUAK1 activity and subcellular localization depend upon protein–protein interactions through the C-terminal domain of the protein. Future investigations will be needed to determine the molecular mechanisms underlying the effects of this ASD-associated NUAK1 STOP mutant.

The behavioral alterations in NUAK1$^{+/-}$ mice are largely consistent with the identification of rare de novo *Nuak1* mutations in autistic patients[17,21]. NUAK1$^{+/-}$ mice display some features associated with the autistic clinical spectrum, including disruption of social novelty preference, moderate cognitive deficits, and a reduction in the PPI reflecting a defect of sensory gating. However, other behavioral alterations typical of ASD were absent in NUAK1$^{+/-}$ mice. In particular, we did not detect stereotyped or repetitive behavior in heterozygous mice, nor any effect on general sociability. In humans, the large heterogeneity in the clinical manifestations characterizing ASD patients partially overlaps clinical manifestations associated with psychiatric disorders such as schizophrenia[44,45].

Importantly most behavioral alterations we report here are equally indicative of ASD, intellectual disability, and schizophrenia-like pathology. Of note the overall activity of NUAK1[+/−] mice was largely normal despite reports linking Nuak1 mutations and AD/HD[22]. One important distinction between ASD and schizophrenia is the age of the onset of phenotypes, schizophrenia being typically diagnosed at adolescence or in young adults following a rather asymptomatic prodromal phase, whereas autism is typically detected in young children. The most commonly used behavioral tests relevant to ASD or schizophrenia in the mouse are performed in young adults. Yet it is worth noting that terminal axon branching defects can be detected as early as P21 in NUAK1[+/−] mice, suggesting that some phenotypic alterations might be present before or around weaning. Future studies might address this question through the measurement of ultrasonic vocalizations or measurement of direct social interactions in juvenile animals. Importantly, we also observed that the branching deficit persisted or even increased up to stage P90, corresponding to the age when mouse behavior was assessed.

Gene-network analyses of genes linked to ASD and other neurodevelopmental disorders identified clusters of candidate genes involved in synapse formation, protein synthesis and RNA translation, signaling pathways, gene regulation, and chromatin remodeling[46–49]. In line with that observation, a significant part of the clinical manifestation can be reversed after the development period, suggesting that synaptic plasticity plays an important role in the disease[50]. Yet in parallel there is ample evidence that ASD and schizophrenia have a strong developmental component and that clinical manifestations may arise from early insults to neural circuits formation. The main consequence of NUAK1 heterozygosity is a reduction in axon branching of cortico-cortical projections, suggesting a direct link between axonal development and neurodevelopmental manifestations. Interestingly a similar reduction in terminal axon branching has been recently observed in a mouse model for the 22q11.2 deletion, strongly associated to schizophrenia[51]. Our work strengthens the link between a disruption of cortical connectivity and disruption in cognitive and social behavior in the mouse and warrants further studies of axon development in the pathophysiology of neurodevelopmental disorders.

A recent study showed that NUAK1 mediates the phosphorylation of the microtubule associated protein TAU on S356 and might be involved in tauopathies such as Alzheimer's disease (Lasagna-Reeves et al., 2016). These results suggest that NUAK1 over-activation might lead to neurodegeneration. In the present study, we have only examined the role of NUAK1 loss-of-function during embryonic and early postnatal development, but since its expression is maintained in the adult brain, future experiments should address NUAK1 function in adult circuits maintenance.

As a whole, we provide evidence that expression of the ASD-associated Nuak1 Q433-Stop mutation disrupts cortical axon branching in vivo. Coupled with our observation that Nuak1 heterozygosity impairs brain development, cortical axon branching in vivo, and that NUAK1[+/−] mice have cognitive and social novelty defects strongly argues that Nuak1 loss-of-function mutations could be causally linked to neurodevelopmental disorders in humans and support targeted efforts to identify additional genetic alterations in the Nuak1 gene. Future experiments will need to determine if other rare de novo loss-of-function mutations identified in genes such as Nuak1 are also haploinsufficient with regard to development of brain connectivity and if this is a general genetic mechanism underlying a fraction of neurodevelopmental and/or neuropsychiatric disorders.

## Methods

**Animals**. Mice breeding and handling was performed according to protocols approved by the Institutional Animal Care and Use Committee at The Scripps Research Institute, Columbia University in New York, and the Ethics committee of the University of Lyon, and in accordance with National Institutes of Health guidelines and the French and European legislation. Time-pregnant females were maintained in a 12 h light/dark cycle and obtained by overnight breeding with males of the same strain. Noon following breeding was considered as E0.5. NUAK1[+/−] mice (Nuak1[tm1Sia]) were obtained from the Riken Institute[20]. Floxed NUAK1 mice were generated from targeted ES cells (Nuak1[tm1a(KOMP)Wtsi]) obtained from the KOMP Repository from UC Davis (https://www.komp.org/index.php, project ID CSD23401). Animals were maintained on a C57Bl/6 J background.

**Image acquisition and analyses**. Confocal images were acquired in 1024 × 1024 mode with a Nikon Ti-E microscope equipped with the C2 (Supplementary Fig. 2 and Fig. 7) or the A1R (all others) laser scanning confocal microscope using the Nikon software NIS-Elements (Nikon). We used the following objective lenses (Nikon): ×10 PlanApo; NA 0.45, ×20 PlanApo VC; NA 0.75, ×60 Apo TIRF; NA 1.49. Time-lapse images were acquired in 512 × 512 mode with a Nikon Ti-E microscope equipped with an EM-CCD Andor iXon3 897 Camera. Mitochondria were imaged at 0.1 frames per second for 30 min. Analysis and tracking of confocal and time-lapse images was performed with NIS-Elements software. Representative neurons were isolated from the rest of the image using ImageJ. Contrast was enhanced and background (autofluorescence of non transfected neurons in culture) removed for better illustration of axons morphology. Kymographs were created with NIS-Elements.

**Behavior analyses**. Two groups of age-related NUAK1 HET mice and WT littermates were submitted to behavioral assays in the Mouse Behavioral Assessment Core at The Scripps Research Institute in San Diego, CA. All quantifications were performed by experienced technicians blind to genotype. Mice were between age 10 and 12 weeks at the start of testing. Procedure, quantification method, order of experiments and group constitution is detailed in the Supplementary methods.

**DNA, plasmids, and cloning**. The empty vector pCAG-IRES-GFP (pCIG2), CRE-expressing vector pCIG2-CRE[52], mVENUS expressing vector pSCV2[53], and mitochondria-tagged DsRed expressing pCAG-mitoDsRED[13] were described previously. peGFP-Flag-mNUAK1 vector was created by amplifying mouse NUAK1 cDNA by PCR and cloning into a peGFP-C1 vector between XhoI and EcoRI sites. A Flag-Tag was added at the N-terminal extremity of the protein by PCR. Kinase-dead (K85A), phosphorylation-defective (T212A), and STOP (Q434STOP) mutants were obtained by site-directed mutagenesis using the Quikchange II site-directed mutagenesis kit from Stratagene using the following primers (forward):

NUAK1 K85A: GCCGAGTGGTTGCTATAGCATCCATCCGTAAGGAC
NUAK1 T212A: GAAGGACAAGTTCTTGCAAGCATTTTGTGGGAGCCCAC
NUAK1 Q434STOP: CCCCTTTCAAAATGGAGTAAGATTTGTGCCGGACTGC.

Neuronal expressing vectors for NUAK1 (pCIG2-Flag-mNUAK1) were obtained by inserting wild type and mutant NUAK1 cDNA into a pCIG2 vector between XhoI and EcoRI sites.

**Ex vivo cortical electroporation**. For the electroporation of dorsal telencephalic progenitors[54], a mix containing 1 µg/µl endotoxin-free plasmid DNA (Midi-prep kit from Macherey-Nagel) plus 0.5% Fast Green (Sigma; 1:20 ratio) was micro-injected in lateral ventricles of the brain of E15.5 embryos using the MicroInject-1000 (BTX) microinjector. Electroporations were performed on the whole head (skin and skull intact) with gold-coated electrodes (GenePads 5 × 7 mm BTX) using an ECM 830 electroporator (BTX) and the following parameters: Five 100 ms long pulses separated by 100 ms long intervals at 20 V. Immediately after electroporation, the brain was extracted and prepared as stated in the neuronal culture section below.

**Primary neuronal culture**. Cortices from E15.5 mouse embryos were dissected in Hank's Buffered Salt Solution (HBSS) supplemented with Hepes (pH 7.4; 2.5 mM), CaCl$_2$ (1 mM, Sigma), MgSO$_4$ (1 mM, Sigma), NaHCO$_3$ (4 mM, Sigma), and D-glucose (30 mM, Sigma), hereafter referred to as cHBSS. Cortices were dissociated in cHBSS containing papain (Worthington) and DNAse I (100 µg/ml, Sigma) for 20 min at 37 °C, washed three times and manually triturated in cHBSS supplemented with DNAse. Cells were then plated at 12.5 × 10$^4$ cells per 35 mm glass bottom dish (Matek) coated with poly-D-lysine (1 mg/ml, Sigma) and cultured for 5 days in Neurobasal medium supplemented with B27 (1×), N2 (1×), L-glutamine (2 mM), and penicillin (5 units/ml)-streptomycin (50 mg/ml). To transfect cultured neurons, we performed magnetofection using NeuroMag (OZ Bioscience) according to the manufacturer's protocol.

**In utero cortical electroporation**. Timed pregnant NUAK1[+/−] or NUAK1[F/F] females were used for In utero cortical electroporation. A mix containing 1 µg/µl endotoxin-free plasmid DNA (Midi-prep kit from Macherey-Nagel) plus 0.5% Fast

Green (Sigma; 1:20 ratio) was injected into one lateral hemisphere of E15.5 embryos using a picospritzer. Electroporation was performed using an ECM 830 electroporator (BTX) with gold-coated electrodes (GenePads 3 × 5 mm BTX) to target cortical progenitors by placing the anode (positively charged electrode) on the side of DNA injection and the cathode on the other side of the head. Four pulses of 45 V for 50 ms with 500 ms interval were used for electroporation. Animals were sacrificed 21 or 90 days after birth by terminal perfusion of 4% paraformaldehyde (PFA, Electron Microscopy Sciences) followed by 2 h post-fixation in 4% PFA.

**Western blotting**. Cells and tissues were lysed in ice-cold lysis buffer containing 25 mM Tris (pH7.5), 2 mM MgCl2, 600 mM NaCl, 2 mM EDTA, 0.5% Nonidet P-40, and 1 × protease and phosphatase mixture inhibitors (Roche). 20 μg aliquots of lysate were separated by electrophoresis on a 10% SDS-polyacrylamide gel and transferred on a polyvinylidene difluoride (PVDF) membrane (Amersham). Incubation with primary antibody was performed overnight in 5% milk-TBS-Tween. The next day, membranes were incubated at room temperature for 1 h with HRP-linked secondary antibodies. Western-blot revelation was performed using ECL (Amersham). Imaging and analysis were performed on a chemidoc imager (Bio-rad) using the ImageQuant software. Images presented in the figures have been cropped around the expected bands for figure design purpose. The original (raw) data for Fig. 1a and Supplementary Figure 1J and including molecular weight markers is presented in Supplementary Figure 10.

**Statistical analyses**. Sample size is detailed in the figures. Statistical tests were performed using Prism (GraphPad Software). All statistical tests are two-tailed. Normality test (D'Agostino & Pierson) was performed prior to analysis to discriminate between parametric or non-parametric assay. Retrospective comparison of variance was performed with a F-test.

## Data availability

Supporting data of this study are available from the corresponding authors on reasonable request.

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

## Acknowledgements

The authors express their gratitude to members of the Polleux and Courchet labs for useful comments and discussion. We are grateful to J. Gogos and A. Diamantopoulou for critical reading of the manuscript. We thank M-E. Mayeur, J. Honnorat, and JF. Ghersi-Egea for assistance for the fear conditioning experiments. This work was supported by NIH-R01NS089456 (F.P.), NIH-F32NS080464 (T.L.), NIH-K99NS091526 (T.L.), the Fondation pour la Recherche Médicale (AJE20141031276) and ERC Starting Grant (678302-NEUROMET). J.C. was the recipient of a grant from the Philippe Foundation.

## Author contributions

V.C., A.J.R., T.L., F.P., and J.C. conceived the experiments and interpreted the results. V.C., P.D.C., T.L., and J.C. performed the experiments. A.R., V.C., G.M.D., and P.D.C. performed behavior analyses. J.C. and F.P. prepared the manuscript.

## Additional information

**Competing interests:** The authors declare no competing interests.

