## [Peer Review File · Nature Communications]

Reviewers' comments:

Reviewer #1 (Remarks to the Author):

Loss-of-function de novo gene mutations identified in ASD patients often occur at a single allele which are predicted to lead to heterozygous individuals. It is important to gain more insights regarding the functional haploinsufficiency in the ASD-related genes. This paper shows that in the ASD gene Nuak1, partial loss of Nuak1 results in significant defects in axon branching and mitochondria dynamics. Importantly, in mice in vivo, loss of Nuak1 at one allele leads to ASD features including impaired social behavior and long-term memory. The study employed comprehensive methodologies including neuron transfection, in utero/ex utero electroporation and transgenic mice. The results are of high quality, solid and convincing. The study, though not entirely new in concept, is nevertheless highly novel regarding the specific gene Nuak1.

1. In many experiments Nuak1 was mutated or gene dose was reduced. However, the protein levels, especially the kinase activity (phosphorylated form), was not examined.

2. NUAK1F/F mice were crossed with NEXCRE mice to drive CRE recombinase expression specifically in neural progenitors of the dorsal telencephalon (Fig 6). Which brain regions are eventually affected? To what extent?

3. Nuak1 STOP caused a loss of protein expression, it is unclear why it could still cause rescue effects (Fig 7).

4. Because Nuak1 belongs to a protein family of multiple members, it is possible that a reduction in Nuak1 results in a compensatory increase in other members or its upstream kinase LKB1, which could complicate the data interpretation.

Minor

1. Fig 4, label I J?

2. There is probably too much discussion on Schizophrenia

Reviewer #2 (Remarks to the Author):

In the present study, the authors characterize mice heterozygous for the NUAK1 gene. Nuak1 haploinsufficiency is demonstrated by marked alterations in gross brain anatomy, mitochondrial motility, in vitro and in vivo dendrite morphology and learning, social and anxiety related behavioral assays. The authors also perform molecular replacement studies in cultured neurons to probe the impact of an ASD related mutation on axonal morphology. While the authors convincingly demonstrate the haploinsufficiency of the NUAK1 gene, the impact of the current work is undercut by a previous publication identifying NUAK1's role in axonal regulation (Courchet et al., Cell, 2013) and a demonstration in the present manuscript that the primary ASD-related mutation in NUAK1 produces a fundamentally different axonal phenotype relative to the loss of one copy of the gene.

Major points:

1) The authors demonstrate with a convincing molecular replacement strategy that the ASD-related mutation in NUAK1 selectively disrupts axon branching without affecting axon length. In contrast, both axon length and axon branching are reduced when NUAK1 expression is reduced in HETs and KO's. This fundamental phenotypic difference between altering gene dose and replacement with the ASD-related truncated form makes it very difficult to judge the disease relevance of the work performed on NUAK1 HETs in the first 6 figures. As a result, the bulk of this work is a demonstration that less severe reductions of NUAK1 expression produce smaller impacts on axon length, branching and mitochondrial migration compared to the NUAK1 KO and Knock Down (Courchet et al., 2013). While the behavioral phenotypes identified in the present work are

novel, their relevance to behavioral phenotypes produced by ASD-related mutations in NUA1, as stated above, is hard to interpret.

2) To test whether the ASD-related NUA1 STOP mutation is a gain of function mutation the authors overexpress the NUA1 STOP in wildtype neurons and observe clear statistically significant reductions in axon branching and branch density compared to wt cortical neurons (Figures 7L&M). Surprisingly, the authors conclude that the "NUA1 STOP mutation causes a loss of function form of the protein" because they "did not observe any marked dominant-negative effect upon NUA1 STOP over-expression in WT cortical neurons". This conclusion is very confusing to this reviewer. The data presented in Figures 7L&M suggest the opposite is the case and is consistent with NUA1 STOP being a dominant negative. Dominant negative characteristics of NUA1 STOP again undercut the utility of a NUA1 HET as an ASD animal model.

3) Mitochondrial motility for the ASD-related mutation is not examined.

4) It would be very reassuring to have a neuronal function assay of some sort to accompany the morphology data (e.g. calcium imaging or electrophysiology) to demonstrate that mutant conditions hinder neuronal communication in some way.

Minor Points:

In figure 4, no results accompany panel indicators "I" and "J".

Reviewer #3 (Remarks to the Author):

This paper demonstrates that the haploinsufficiency of NUA1, an autism candidate gene, leads to defects in cortical development as well as impairments in behaviors associated with ASD, schizophrenia, and ID in mice. NUA1 haploinsufficient mice displayed the enlargement of lateral ventricles, thinning of the cortex, and dysfunction in cortical axon branching. Behavioral defects exhibited by NUA1-haploinsufficient mice include long-term spatial memory impairment, social novelty deficit, sensorimotor gating abnormality, and reduced prepulse inhibition. The levels of behavioral impairment such as postnatal growth retardation and defects in acoustic startle response were to some extent dependent on the sex of the mice. The authors have also generated constructs that have ASD-associated premature termination codon mutation (NUA1 STOP), kinase-dead K85A mutation, or phosphor-defective T212A mutation, and performed rescue experiments and dominant negative test in NUA1 null and wild-type neurons, respectively. The results suggest that NUA1 kinase activity is important for axonal growth and branching, while CTD may contribute only to axonal branching. The authors conclude that NUA1 STOP truncation has a dominant negative effect on collateral branch formation.

This paper provides in vivo functions of NUA1 in the regulation of brain development, axon branching, and several behaviors. The experiments are well designed, and the results are properly interpreted. Data from cell biological experiments and behavioral experiments are presented in good balance.

Major comments:

1. The Barnes maze memory impaired one month after the acquisition seems to be a mild phenotype. The authors need to test other forms of learning and memory such as contextual fear conditioning.
2. One of the major phenotypes of ASD is repetitive behavior. The authors no significant repetitive behavior (grooming) in an open-field context (Figure S5). But the results could be confounded by a novel environment. I recommend to measure repetitive behaviors such as self-grooming and digging in home-cage environments.
3. Another important behavior associated with ASD is ultrasonic vocalizations (USVs). The authors may want to consider performing USV measurements in pups and/or adult mice.

4. The larger size form of NUAK1 protein (upper band in Western blot; Figure S1J) seems to appear in later stages of development (after P21) while the smaller size form seems to be present event at P1. According to Figure 1, the upper band of NUAK1 protein seems to be significantly decreased in the cortical neurons of HET mice and KO mice while the lower band seems to remain constant or even increase. Could this perhaps explain why the cortical development in embryonic brain seems normal? In addition, it is unclear why the relative amounts of the upper and lower bands are different in Figure 1 and Figure S1.

5. In Figure 7 A, the NUAK1 STOP protein still has the kinase domain, and this protein is not degraded (Figure 7B). Does the NUAK1 STOP protein have intact kinase activity. If so, is the rescue of axon length by the expression of NUAK1 STOP (Figure 7 H and K) attributable to the kinase activity?

Minor comments:

1. It is interesting that only contralateral terminal axon branching is affected in Nuak1-deficient mice. Known or potential mechanisms should be discussed.

2. The authors' previous study (ref. 13) and present study consistently show changes in mitochondria trafficking. Known or potential mechanisms should be discussed

3. The authors conclude that there's no difference between WT, HET, and KO embryo in cortical lamination (Figure S2D-F). However, TBR1-positive cells in KO mice (Figure S2E) seem to be different from those in WT and HT samples. This should be clarified.

4. Figure 4: the bar graph for prepulse inhibition should be labeled Figure 4I. There is a label for Figure 4J but no corresponding figure. Figure 4J is not mentioned in the main text.

5. The authors point to Figure S2 for histochemical analysis of P21 and P40 mice, but the data for P21 is missing.

6. Figure S2 I&K typo: there is a space between NUA and K+/+.

Rebuttal NCOMMS-18-03986

First, we would like to thank the reviewers for their overall level of enthusiasm towards our results and for their constructive suggestions. We have addressed most of their points with new data, or extensively modified the text per their suggestions. We are convinced that the revised manuscript is significantly improved as a result.

Reviewers' comments:

Reviewer #1 (Remarks to the Author):

Loss-of-function de novo gene mutations identified in ASD patients often occur at a single allele which are predicted to lead to heterozygous individuals. It is important to gain more insights regarding the functional haploinsufficiency in the ASD-related genes. This paper shows that in the ASD gene *Nuak1*, partial loss of *Nuak1* results in significant defects in axon branching and mitochondria dynamics. Importantly, in mice *in vivo*, loss of *Nuak1* at one allele leads to ASD features including impaired social behavior and long-term memory. The study employed comprehensive methodologies including neuron transfection, *in utero/ex utero* electroporation and transgenic mice. The results are of high quality, solid and convincing. The study, though not entirely new in concept, is nevertheless highly novel regarding the specific gene *Nuak1*.

1. In many experiments *Nuak1* was mutated or gene dose was reduced. However, the protein levels, especially the kinase activity (phosphorylated form), was not examined.

Response:

Currently there are no commercially available antibodies directed toward the T212 phosphorylation site for NUAK1. Recent papers have used the phosphorylation of MYPT1, a direct albeit non-specific target of NUAK1, as an indirect readout of the kinase activity of the protein^{1,2}. We obtained this antibody from the MRC-PPU at Dundee and analyzed it by Western-blot on neuronal lysates. However, we do not detect any consistent signal on our Western-blots and thus can not reliably use this strategy as a way to measure NUAK1 kinase activity.

Previously, we found a dose reduction of NUAK1 protein expression in *NUAK1*^{+/-} neurons³ and observe the same in this manuscript (Figure 1A). Furthermore, we observed an increase in NUAK1 protein expression in HEK cells upon overexpression of the NUAK1 activator LKB1, and conversely a decrease in NUAK1 protein expression upon knockout of LKB1 in cortical neurons³. Similar observations have been reported by Dario Alessi's group in MEF and Hela cells (see Figure 4B and 4D from¹). All together these results suggest that NUAK1 activity and protein expression or stability are directly correlated. Thus the reduction in NUAK1 protein expression is likely accompanied by a corresponding reduction in NUAK1 kinase activity.

Finally, we performed an *in vitro* kinase assay following expression of wild-type and mutated NUAK1 in HEK cells and immunoprecipitation. Using this strategy we were able to detect the specific activity of NUAK1 towards a target peptide. As predicted the catalytic activity of NUAK1 was virtually abolished by the kinase domain (K85A) mutation. However the catalytic activity of the NUAK1 STOP mutant was largely unaffected *in vitro*. The fact that NUAK1 STOP mutant displays catalytic activity *in vitro* explains how it may partially rescue the axonal phenotype resulting from NUAK1 deletion. As discussed in the manuscript, the truncation of a large fraction of the C-terminus end of NUAK1 might impair protein-protein interactions involved in the regulation of NUAK1 location or function. Future studies will need to clarify the mechanism involved in the loss of axon branches but we strongly feel that these experiments are beyond the scope of the present study.

2. *NUAK1*F/F mice were crossed with NEXCRE mice to drive CRE recombinase expression

specifically in neural progenitors of the dorsal telencephalon (Fig 6). Which brain regions are eventually affected? To what extent?

Response:

The NEX^{CRE} mice have been characterized previously⁴. The pattern of CRE expression has been extensively described and can be found as table 1 in Goebbels *et al*, 2006⁴. LacZ reporter gene expression experiments demonstrated that CRE is strongly expressed in the dorsal telencephalon in the developing and adult brain, including regions of the cortex, hippocampus and amygdala. In the adult, CRE expression has also been detected in discrete regions of the hypothalamus, and nuclei of the midbrain, hindbrain and cerebellum.

Importantly, this driver does not drive CRE expression in neural progenitors in the cerebral cortex as demonstrated by the lack of CRE expression in the Ventricular Zone (VZ). Thus CRE expression is restricted to Intermediate Progenitors (IP) and post-mitotic glutamatergic excitatory neurons, and excluding GABAergic interneurons and later-born glial cells (such as astrocytes).

We edited the results section to better explain the pattern of expression of CRE using the NEX driver.

3. Nuak1 STOP caused a loss of protein expression, it is unclear why it could still cause rescue effects (Fig 7).

Response:

We show (Figure 7B) that NUA1 STOP mutant can be detected by Western-blot upon expression in HEK cells, suggesting that the stability of the protein is not affected by the mutation. As discussed above, we now add evidence that NUA1 STOP mutant has an intact catalytic activity using an in vitro kinase assay (Figure 7C). We believe this explains how this mutant can have a partial rescue effect on axon elongation. The new experiments (kinase assay) and the potential mechanisms by which NUA1 STOP can still have some rescue activity are now added to the manuscript.

4. Because Nuak1 belongs to a protein family of multiple members, it is possible that a reduction in Nuak1 results in a compensatory increase in other members or its upstream kinase LKB1, which could complicate the data interpretation.

Response:

In this study, we report a dose-dependent effect of NUA1 loss on mitochondria trafficking, axon outgrowth and branching. We furthermore measured behavioral alterations upon loss of one copy of NUA1 that are consistent with the phenotype of a complete KO. These observations suggest that NUA1 is haploinsufficient for the above-mentioned phenotypes and thus a lack of compensatory mechanism by other kinases of the AMPK family.

Following the suggestion of this reviewer, we determined the expression of several members of the AMPK family in NUA1^{+/-} and NUA1^{-/-} neuronal cultures. This new result is incorporated in Supplementary Figure 2M. As previously observed³, NUA2, the closest NUA1 related kinase, is virtually absent in postmitotic cortical neurons. We also assessed AMPK expression, since studies in cancer cell lines have suggested that NUA1 may regulate the stability of the AMPK complex^{2,5}. Using an antibody directed toward the two catalytic isoforms of AMPK (α 1 and α 2), we did not observe any change in AMPK protein quantity in NUA1 HET and KO neuronal cultures. Finally, we used antibodies toward BRSK1 (SAD-B) and BRSK2 (SAD-A), as these two kinases are involved in axon specification^{6,7}. BRSK2 expression was unaffected in NUA1 HET and KO samples. However we observed a dose-dependent upregulation of BRSK1 protein expression in NUA1 HET and KO samples. The functional relevance of this observation is however unclear, since we previously demonstrated that overexpression of other AMPK-related kinases NUA2, BRSK1 (SAD-B) and AMPK α 2 have no significant effect on axon length

and branching³.

Minor

1. Fig 4, label I J?

Response:

We thank the reviewer for noticing this typo in the figures. This has been corrected in the present version of the manuscript.

2. There is probably too much discussion on Schizophrenia

Response:

The discussion of the paper has been modified to include the new results and the comments of the three reviewers.

Reviewer #2 (Remarks to the Author):

In the present study, the authors characterize mice heterozygous for the *NUAK1* gene. *Nuak1* haploinsufficiency is demonstrated by marked alterations in gross brain anatomy, mitochondrial motility, in vitro and in vivo dendrite morphology and learning, social and anxiety related behavioral assays. The authors also perform molecular replacement studies in cultured neurons to probe the impact of an ASD related mutation on axonal morphology. While the authors convincingly demonstrate the haploinsufficiency of the *NUAK1* gene, the impact of the current work is undercut by a previous publication identifying *NUAK1*'s role in axonal regulation (Courchet et al., Cell, 2013) and a demonstration in the present manuscript that the primary ASD-related mutation in *NUAK1* produces a fundamentally different axonal phenotype relative to the loss of one copy of the gene.

Major points:

1) The authors demonstrate with a convincing molecular replacement strategy that the ASD-related mutation in *NUAK1* selectively disrupts axon branching without affecting axon length. In contrast, both axon length and axon branching are reduced when *NUAK1* expression is reduced in HETs and KO's. This fundamental phenotypic difference between altering gene dose and replacement with the ASD-related truncated form makes it very difficult to judge the disease relevance of the work performed on *NUAK1* HETs in the first 6 figures. As a result, the bulk of this work is a demonstration that less severe reductions of *NUAK1* expression produce smaller impacts on axon length, branching and mitochondrial migration compared to the *NUAK1* KO and Knock Down (Courchet et al., 2013). While the behavioral phenotypes identified in the present work are novel, their relevance to behavioral phenotypes produced by ASD-related mutations in *NUAK1*, as stated above, is hard to interpret.

Response:

We thank the reviewer for his critical reading of the manuscript. The reviewer rightly notes that a major phenotypical difference between the genetic knock-out approach (mimicking the effect of a loss of one copy of *NUAK1*) and the expression of the mutant protein is the decoupling between axon elongation and collateral branching. However these two models are not mutually exclusive:

1) The main goal of our paper was to explore the functional consequence of *de novo* mutations affecting *Nuak1* and that are presumably somatic, heterozygous mutations. In this context the study of two genetic models (*NUAK1* HET and *NUAK1* conditional KO) demonstrates that *Nuak1* is haploinsufficient with regard to its role during the development of mouse cortical connectivity. This observation is novel and bears relevance to any mutation found in the *Nuak1* gene in a pathological context and presumably inactivating the gene.

2) As discussed below, the interpretation of the NUAK1 STOP mutation phenotype is complex since we had to distinguish any effect resulting from the loss of function of the protein and a potential dominant-negative effect. The gene-replacement strategy is a quick way to assess the functional relevance of mutations found in *Nuak1*. Using this strategy we observed that expression of the ASD-associated NUAK1 STOP mutant fails to rescue axon branching. This result demonstrates that this mutation disrupts NUAK1's function, despite the conserved kinase activity (new result added to Figure 7C). However, this assay was performed in a null background and with a CAG-driven overexpression of the mutant protein, which does not faithfully recapitulate the human pathological context. The only strategy to directly address the disease-relevance of the NUAK1 STOP mutation would be to generate a CRISPR mouse line recapitulating the mutation within the *Nuak1* locus. This would require a significant amount of time and effort which we feel is outside the scope of the present study, but agree would be an important future direction.

3) We observed throughout this manuscript as well as our previous study that NUAK1 exerts two distinct functions on axon morphogenesis, namely a control of axon elongation and terminal axon branching. Upon knock-down or knock-out of NUAK1, we observe *in vitro* a reduction of the axon's length as well as a reduction of collateral branching. *In vivo*, the most obvious phenotype is a reduction in terminal axon branching. Importantly, we reported previously that LKB1 knockout impaired axon elongation as well *in vivo* and that axons reach the contralateral hemisphere later than in control condition. However they do ultimately reach their final destination. To what extent this delay in reaching their destination contributes to the branching defect *in vivo* is still an open question.

2) To test whether the ASD-associated NUAK1 STOP mutation is a gain of function mutation the authors overexpress the NUAK1 STOP in wildtype neurons and observe clear statistically significant reductions in axon branching and branch density compared to wt cortical neurons (Figures 7L&M). Surprisingly, the authors conclude that the "NUAK1 STOP mutation causes a loss of function form of the protein" because they "did not observe any marked dominant-negative effect upon NUAK1 STOP over-expression in WT cortical neurons". This conclusion is very confusing to this reviewer. The data presented in Figures 7L&M suggest the opposite is the case and is consistent with NUAK1 STOP being a dominant negative. Dominant negative characteristics of NUAK1 STOP again undercut the utility of a NUAK1 HET as an ASD animal model.

Response:

Below is a comparison of the average \pm 95% CI for data presented in Figure 7N (branch per 250 μ m).

	Control	CRE	CRE + WT	CRE + KD	CRE + TA	CRE + STOP	Control + WT	Control + STOP
Mean	2,678	1,669	4,547	1,729	1,814	1,374	3,623	2,159
\pm 95% CI of mean	0,342	0,301	0,808	0,32	0,29	0,273	0,497	0,341
Mean (%)	100%	62%	170%	65%	68%	51%	135%	81%
\pm 95% CI of mean (%)	13%	11%	30%	12%	11%	10%	19%	13%

This reviewer correctly notes that overexpression of NUAK1 STOP mutant in wild-type neurons does have a statistically significant impact over axon branching. However the reduction in axon branching is only limited compared to a true loss of function of the kinase function, such as that can be achieved by the knockout. Overexpression of NUAK1 STOP mutant reduced axon branching to 2,159 \pm 0,341 branch per 250 μ m (from 2,678 \pm 0,342 in the control condition), which is a 19% reduction of branch number on average, and not as drastic as the 38% reduction observed upon CRE mediated inactivation of NUAK1.

Furthermore, whereas NUAK1 inactivation has a marked effect on axon elongation, the overexpression of NUAK1 STOP mutant did not reduce the maximal axon length (Figure 7L). Finally NUAK1 STOP mutant overexpression had no marked effect on mitochondria trafficking in wild-type neurons (Supplementary Figure 9).

Taken together, our results show that NUAK1 STOP mutant has only a limited functional

impact over the endogenously expressed NUA1 protein and thus behaves mostly as a loss of function mutant phenocopying haploinsufficiency. We clarified our interpretation and discuss it more in the manuscript.

3) Mitochondrial motility for the ASD-related mutation is not examined.

Response:

The corresponding experiment has been performed and included in Supplementary figure 9. The main result confirms the observation for axon morphogenesis, *ie.* NUA1 STOP mutant fails to rescue the mitochondrial phenotype in NUA1 deficient neurons. Importantly, overexpression of STOP mutant in wild-type neurons had no detectable effect on mitochondria motility, thus reinforcing our interpretation that NUA1 STOP mutant mostly behaves like a loss of function allele.

4) It would be very reassuring to have a neuronal function assay of some sort to accompany the morphology data (e.g. calcium imaging or electrophysiology) to demonstrate that mutant conditions hinder neuronal communication in some way.

Response:

Our manuscript aimed at testing the hypothesis that NUA1 is haploinsufficient for brain development and therefore that an alteration of NUA1 expression affects the development of cortical connectivity. We did indeed observe functional deficits in the behavioral experiments that to our opinion constitute a clear indication that circuit function is altered upon NUA1 dose reduction. Whether the pattern of circuits or cortical circuit function are affected remains unknown at this stage, however we feel these experiments would by themselves constitute an entire study and therefore they fall out of the scope of the present paper.

Minor Points:

In figure 4, no results accompany panel indicators "I" and "J".

Response:

As for reviewer 1 we thank this reviewer for his careful reading and noticing this typo that has been corrected in the revised version of the manuscript.

Reviewer #3 (Remarks to the Author):

This paper demonstrates that the haploinsufficiency of NUA1, an autism candidate gene, leads to defects in cortical development as well as impairments in behaviors associated with ASD, schizophrenia, and ID in mice. NUA1 haploinsufficient mice displayed the enlargement of lateral ventricles, thinning of the cortex, and dysfunction in cortical axon branching. Behavioral defects exhibited by NUA1-haploinsufficient mice include long-term spatial memory impairment, social novelty deficit, sensorimotor gating abnormality, and reduced prepulse inhibition. The levels of behavioral impairment such as postnatal growth retardation and defects in acoustic startle response were to some extent dependent on the sex of the mice. The authors have also generated constructs that have ASD-associated premature termination codon mutation (NUA1 STOP), kinase-dead K85A mutation, or phosphor-defective T212A mutation, and performed rescue experiments and dominant negative test in NUA1 null and wild-type neurons, respectively. The results suggest that NUA1 kinase activity is important for axonal growth and branching, while CTD may contribute only to axonal branching. The authors conclude that NUA1 STOP truncation has a dominant negative effect on collateral branch formation.

This paper provides *in vivo* functions of NUA1 in the regulation of brain development, axon branching, and several behaviors. The experiments are well designed, and the results are properly interpreted. Data from cell biological experiments and behavioral experiments are

presented in good balance.

Major comments:

1. The Barnes maze memory impaired one month after the acquisition seems to be a mild phenotype. The authors need to test other forms of learning and memory such as contextual fear conditioning.

Response:

We followed this reviewer's advice and performed Cued and Contextual Fear Conditioning (FC) in NUAK1^{+/-} and in NUAK1 cKO mice. For constraints of time we only tested cohorts of female animals. We observed a similar effect in both mice lines and these results have been included to the manuscript. Briefly, all groups of mice behaved the same during the initial training period regardless of their genotype. On day 2, we detected a marked decrease of the cued fear response (total number and duration of startle events, defined as a total absence of movement for >2 seconds and measured automatically using the Ethovision software) in NUAK1^{+/-} and in NUAK1 cKO mice alike. Interestingly the response to contextual fear conditioning was reduced in NUAK1^{+/-} and in NUAK1 cKO mice but the differences with the control groups is not significant, suggesting the neural circuits underlying the cued fear conditioning may have a more pronounced sensitivity to the inactivation of NUAK1.

Taken together, this new data point to a deficit in associative memory and strengthen our conclusion that cognitive processes are impaired upon *Nuak1* heterozygosity.

2. One of the major phenotypes of ASD is repetitive behavior. The authors no significant repetitive behavior (grooming) in an open-field context (Figure S5). But the results could be confounded by a novel environment. I recommend to measure repetitive behaviors such as self-grooming and digging in home-cage environments.

Response:

We agree with this reviewer that the open field assay is a stressful environment for the mice. We performed home-cage observations of NUAK1^{+/-} mice and measured spontaneous behaviors: grooming, digging (total time and frequency), rearing and scratching (frequency).

Overall our results were largely in range with the results of the open field, with little differences in spontaneous repetitive behaviors such as grooming or rearing. We noted however a mild increase in the level of activity of NUAK1^{+/-} mice that can be measured especially through the increased digging time. This difference was only moderate and we believe did not impact our interpretation of other behavioral assays. Importantly we went back to the analysis for the data in the 3-chamber social assay and there was no difference in the number of doors passed between NUAK1 cKO, cHET and WT animals. Furthermore NUAK1^{+/-} and NUAK1 cKO mice had a level of activity comparable to WT littermates during the training phase of the FC assay (day 1, Fig 3G, Fig 6D).

We added this new result to the manuscript and discuss a potential environment effect within the open field assay.

3. Another important behavior associated with ASD is ultrasonic vocalizations (USVs). The authors may want to consider performing USV measurements in pups and/or adult mice.

Response:

We thank this reviewer for this suggestion. For a number of reasons, it has not been feasible to perform these experiments via collaboration. We discuss in the manuscript the benefit of such an assay.

Furthermore we performed some measurement of social interaction in juvenile mice. The results suggest an absence of difference between NUAK1^{+/-} and control littermates. Yet because this assay has to be performed between similar aged mice and between mice that

have never been in contact (*ie* excluding littermates), the number of animals we could test is actually low and for this reason the data was not included to the manuscript.

4. The larger size form of NUA1 protein (upper band in Western blot; Figure S1J) seems to appear in later stages of development (after P21) while the smaller size form seems to be present event at P1. According to Figure 1, the upper band of NUA1 protein seems to be significantly decreased in the cortical neurons of HET mice and KO mice while the lower band seems to remain constant or even increase. Could this perhaps explain why the cortical development in embryonic brain seems normal? In addition, it is unclear why the relative amounts of the upper and lower bands are different in Figure 1 and Figure S1.

Response:

The major band in mouse brain samples with anti-NUAK1 antibody was detected at 75kDa, which is consistent with the size of the protein (658 aminoacids and theoretical mass 73.661 kDa according to Uniprot – accession number Q641K5). We indeed observed a slower migrating band in the older animals (P21 and Adults samples) that was not detected in early postnatal samples. The significance of this band is not known. Interestingly we could also detect two bands upon overexpression of NUA1 in HEK293T cells (Figure 7B) and the upper band was decreased, but not abolished, in the T212A mutant sample, suggesting it could correspond to some post-translational modification of NUA1.

However in most experiments performed in primary neuronal cultures as is the case for Figure 1A, we only observed the major, 75kDa band and not the upper band. Importantly the lower band detected in Figure 1A is a non-specific band detected in some primary neuronal cultures. Indeed this band is still present in the KO sample and its size is less than 75kDa, suggesting it corresponds to another protein.

For clarity, we added on Figure 1 a note that the lower band is a non specific band and provide to the reviewer the raw blot data showing the difference in band size (see annex at the end of this document).

5. In Figure 7 A, the NUA1 STOP protein still has the kinase domain, and this protein is not degraded (Figure 7B). Does the NUA1 STOP protein have intact kinase activity. If so, is the rescue of axon length by the expression of NUA1 STOP (Figure 7 H and K) attributable to the kinase activity?

As stated above, we assessed the kinase activity of the NUA1 STOP mutant and could confirm that its catalytic activity is similar to the activity of full-length NUA1 in vitro against a target peptide. This data explains how the mutant can rescue axon growth in a null background, whereas a kinase-dead form of NUA1 cannot.

Minor comments:

1. It is interesting that only contralateral terminal axon branching is affected in Nuak1-deficient mice. Known or potential mechanisms should be discussed.

Response:

We agree that this is an interesting phenotype and further discussed the potential differences in the molecular and cellular mechanisms underlying ipsilateral versus contralateral branching.

2. The authors' previous study (ref. 13) and present study consistently show changes in mitochondria trafficking. Known or potential mechanisms should be discussed

Response:

We assessed changes in axonal trafficking of mitochondria as a direct consequence of

NUAK1 function and how it is mechanistically linked to axon development. However the present study did not aim to and does not provide any significant mechanistic insight into how the LKB1-NUAK1 signaling pathway controls mitochondria immobilization in the axon. Importantly the mechanism tying NUAK1 to mitochondria motility and presynaptic capture is still poorly understood but might involve a protein called syntaphilin, as shown in our 2013 Cell paper. This important topic is currently being investigated in our labs and other labs but we feel that a discussion of this specific mechanism in this manuscript is not justified, since we do not have additional insights to provide compared to the discussion in the 2013 Cell paper.

3. The authors conclude that there's no difference between WT, HET, and KO embryo in cortical lamination (Figure S2D-F). However, TBR1-positive cells in KO mice (Figure S2E) seem to be different from those in WT and HT samples. This should be clarified.

Response:

This histochemical analysis in embryonic brains has been performed several time and we did not detect consistent alteration in neuronal layers formation. We performed another batch of histochemistry and provide new and less ambiguous images that we believe better represent a typical experiment.

4. Figure 4: the bar graph for prepulse inhibition should be labeled Figure 4I. There is a label for Figure 4J but no corresponding figure. Figure 4J is not mentioned in the main text.

Response:

As mentioned above this typo resulting from a change in figure organization has been corrected in the revised version of the manuscript.

5. The authors point to Figure S2 for histochemical analysis of P21 and P40 mice, but the data for P21 is missing.

Response:

The data at P21 correspond to the analysis of brain region size from Figure 1D-E. Histochemical analyses of neuronal layers and soma-axon markers were performed on P40 brains. Furthermore the enlargement of the ventricles is visible at P21 (Figure 1D-E) and P40 (Supplementary figure 2). We modified the text in the results sections to alleviate this ambiguity.

6. Figure S2 I&K typo: there is a space between NUA and K+/>+.

Response:

We corrected the typo and thank the reviewer for his careful reading of the manuscript.

References cited

1. Zagórska, A. *et al.* New roles for the LKB1-NUAK pathway in controlling myosin phosphatase complexes and cell adhesion. *Sci Signal* **3**, ra25 (2010).
2. Monteverde, T. *et al.* Calcium signalling links MYC to NUAK1. *Oncogene* **268**, 9194 (2017).
3. Courchet, J. *et al.* Terminal Axon Branching Is Regulated by the LKB1-NUAK1 Kinase Pathway via Presynaptic Mitochondrial Capture. *Cell* **153**, 1510–1525 (2013).
4. Goebbels, S. *et al.* Genetic targeting of principal neurons in neocortex and hippocampus of NEX-Cre mice. *Genesis* **44**, 611–621 (2006).
5. Liu, L. *et al.* Deregulated MYC expression induces dependence upon AMPK-related kinase 5. *Nature* **483**, 608–612 (2012).
6. Kishi, M., Pan, Y. A., Crump, J. G. & Sanes, J. R. Mammalian SAD kinases are required for

- neuronal polarization. *Science* **307**, 929–932 (2005).
7. Barnes, A. P. *et al.* LKB1 and SAD kinases define a pathway required for the polarization of cortical neurons. *Cell* **129**, 549–563 (2007).

Annex: supplementary figure for reviewer #1

Raw data (blots) from Figure 1A

10% SDS-PAGE - Protein Ladder Dual Color (BioRad)

Raw data (blots) from Supplementary figure 1J

10% SDS-PAGE - Protein Ladder Dual Color (BioRad)

REVIEWERS' COMMENTS:

Reviewer #1 (Remarks to the Author):

My previous concerns have been well addressed in the revised manuscript. The current version merits considered for publication.

Reviewer #2 (Remarks to the Author):

The authors have addressed by concerns. I think the manuscript is fine in its present form.

Reviewer #3 (Remarks to the Author):

The authors have fully addressed all of my comments. I do not have any further comments.

Point by point response for manuscript NCOMMS-18-03986A

We would like to thank the reviewers and editors for the handling and critical reading of this manuscript. Following the editorial requests we performed the modifications of the manuscript. The changes have been made using the track change function of MsWord.

REVIEWERS' COMMENTS:

Reviewer #1 (Remarks to the Author):

My previous concerns have been well addressed in the revised manuscript. The current version merits considered for publication.

Reviewer #2 (Remarks to the Author):

The authors have addressed by concerns. I think the manuscript is fine in its present form.

Reviewer #3 (Remarks to the Author):

The authors have fully addressed all of my comments. I do not have any further comments.

Response:

We thank the reviewers for their constructive comments during the revision of this manuscript and appreciate their enthusiasm for this work.